# Coevolution of the *Tlx* homeobox gene with medusa development (Cnidaria: Medusozoa)

Matthew Travert [1✉], Reed Boohar[1], Steven M. Sanders [2], Manon Boosten[3,4], Lucas Leclère[3,5], Robert E. Steele[6] & Paulyn Cartwright [1]

Cnidarians display a wide diversity of life cycles. Among the main cnidarian clades, only Medusozoa possesses a swimming life cycle stage called the medusa, alternating with a benthic polyp stage. The medusa stage was repeatedly lost during medusozoan evolution, notably in the most diverse medusozoan class, Hydrozoa. Here, we show that the presence of the homeobox gene *Tlx* in Cnidaria is correlated with the presence of the medusa stage, the gene having been lost in clades that ancestrally lack a medusa (anthozoans, endocnidozoans) and in medusozoans that secondarily lost the medusa stage. Our characterization of *Tlx* expression indicate an upregulation of *Tlx* during medusa development in three distantly related medusozoans, and spatially restricted expression patterns in developing medusae in two distantly related species, the hydrozoan *Podocoryna carnea* and the scyphozoan *Pelagia noctiluca*. These results suggest that *Tlx* plays a key role in medusa development and that the loss of this gene is likely linked to the repeated loss of the medusa life cycle stage in the evolution of Hydrozoa.

[1] Department of Ecology and Evolutionary Biology, University of Kansas, Lawrence, KS, USA. [2] Department of Surgery, Thomas E. Starzl Transplantation Institute, University of Pittsburgh, Pittsburgh, PA, USA. [3] Sorbonne Université, CNRS, Laboratoire de Biologie du Développement de Villefranche-sur-Mer (LBDV), Villefranche-sur-Mer, France. [4] Sorbonne Université, CNRS, Laboratoire d'Océanographie de Villefranche, Villefranche-sur-Mer, France. [5] Sorbonne Université, CNRS, Biologie Intégrative des Organismes Marins, BIOM, Banyuls-sur-Mer, France. [6] Department of Biological Chemistry, University of California, Irvine, CA, USA. ✉email: travert.matthew@gmail.com

Medusae (=jellyfish) represent part of a complex life cycle that is characteristic of the cnidarian subphylum Medusozoa, which includes true jellyfish (Scyphozoa), box jellyfish (Cubozoa), stalked jellyfish (Staurozoa) and hydromedusae (Hydrozoa). Medusozoans possess a metagenetic life cycle alternating between an asexual phase in the form of a sessile polyp and a sexually reproducing, typically pelagic phase called the medusa or jellyfish. The medusa exhibits several distinct features including a bell-shaped morphology, striated muscles, gonads, and sensory organs. The parasitic Endocnidozoa is sister to Medusozoa[1] and lacks a definitive polyp or medusa stage. The other major cnidarian subphylum, Anthozoa (corals, anemones, and sea pens) also lacks a medusa as well as all traits associated with this free-living stage, and instead possesses a monogenetic life cycle, comprising only sexually and asexually reproducing sessile polyps.

Despite the medusa being characteristic of Medusozoa, losses of this life cycle stage within Hydrozoa were frequent and most likely due to developmental heterochrony[2–4]. The fully developed hydromedusa exhibits distinctive features such as a muscular structure at the bell margin called a velum, striated muscles, marginal tentacles and tentacle bulbs, a manubrium which contains the gut and mouth, and a gastrovascular system composed of radial and circular canals. The truncation of the medusa stage was shown to correlate with the loss or reduction of the aforementioned features, with the varying degrees of medusa truncation often mirroring stages of medusa development[5–8]. The differing degrees of medusa truncation across species can be thought as a type of paedomorphic progenesis[9,10], where somatic development is truncated due to early sexual maturation.

While the development of the hydrozoan polyp has been well studied, particularly in the model systems *Hydra*[11] and *Hydractinia*[12], the molecular mechanisms underlying the development of the hydromedusa remain poorly understood. While some studies have indicated that medusa development co-opts key developmental pathways functioning in the polyp[13–17], recent studies suggest that some medusa-specific transcription factors might act as molecular switches that regulate aspects of medusa development[18,19]. Among identified medusa-specific genes (gene whose expression is restricted to the medusa stage) a large proportion are homeobox genes, suggesting that these genes may play a key role in the development and maintenance of the medusa.

One purported medusa-specific gene is the T Cell Leukemia Homeobox gene (*Tlx*)[18]. In vertebrates, *Tlx* (also referred to as *HOX11*, despite being part of the NK subclass) is involved in spleen organogenesis, brain, and skeleton patterning and in *Drosophila melanogaster* it is involved in distal patterning of the leg *(clawless)*[20,21]. *Tlx* could not be found in the genome of the sea anemone *Nematostella vectensis*[22] and thus was originally thought to be absent in cnidarians. Here we show that *Tlx* is indeed present in cnidarians. However, our survey of this gene across cnidarian genomes found that a *Tlx* gene is present only in those cnidarian lineages exhibiting a medusa stage. *Tlx* is absent in all anthozoans and endocnidozoan genomes surveyed. In addition, an intact *Tlx* gene is absent in nearly all hydrozoans surveyed that have lost a medusa life cycle stage. In several distantly related medusae-bearing species, we found that *Tlx* expression is upregulated during medusa development and its expression is consistent with it playing a role in medusa patterning.

## Results

### The cnidarian *Tlx* ortholog shares a highly conserved genomic structure with bilaterian *Tlx*. *Tlx* is part of the NK-L subclass of homeobox genes. The TLX protein has an EH1 domain near the

N-terminus, a homeodomain, and a signature motif, RRIGHPY just upstream of the homeodomain, called the N-terminal arm (Fig. 1a). Although the N-terminal arm is highly conserved, the third residue varies in medusozoans with the following frequencies found in surveyed species, isoleucine (~70%), leucine (~17%) and valine (~12%). In surveyed bilaterians, the third residue was consistently found to be either an isoleucine (~88%) or a valine (~12%). In our search of public databases for *Tlx*, the EH1 domain, as well as the N-terminal arm were invariably found in complete sequences of *Tlx* in both cnidarians and bilaterians. These conserved regions could not be found in the putative *Tlx* orthologs of sponges and placozoans, and although ctenophore putative *Tlx* orthologs had an EH1 domain, they lacked the N-terminal arm. In addition to low sequence identity in the homeodomain, a Bayesian phylogenetic analysis did not recover the putative *Tlx-like* gene previously identified in ctenophores[23,24], within the strongly supported bilaterian and cnidarian *Tlx* orthology group (Supplementary Fig. S1). Although Ryan et al.[25] reported a *Tlx* gene in the sea anemone *Nematostella*, this gene lacks the signature EH1 domain and N-terminal arm and was not recovered in the *Tlx* orthology group with sufficient support in their analysis. In our analyses, we recovered this sequence as part of a well-supported orthology group (BS = 70, PP = 0.99) comprising NK-L genes from anthozoans and hydrozoans and sponges (Supplementary Fig. S2), all of which lack the EH1 domain and N-terminal arm and is herein referred to as the *Tlx-like* gene. The *Nematostella vectensis Tlx-like* gene aligned with *Tlx* of cnidarians and bilaterians is shown in Supplementary Fig. S3. Vertebrates possess three *Tlx* paralogs all possessing the EH1 domain and the N-terminal arm, suggesting two duplication events in the last common ancestor of vertebrates. A phylogenetic tree consisting of 38 cnidarian taxa and 11 vertebrate taxa including their three paralogs is shown in Fig. 1b. *Tlx* forms a well-supported orthology group (BS = 83, PP = 0.99). Although vertebrate *Tlx* paralogy groups are respectively well supported, a relationship between cnidarian *Tlx* and a specific vertebrate paralog could not be recovered (TLX1/3) with sufficient support. A phylogenetic analysis including select protostome, cnidarian and vertebrate *Tlx* genes also formed a well-supported *Tlx* orthology group in the Bayesian analysis (PP = 0.98) but failed to recover specific orthology relationships between major taxa (Supplementary Fig. S1).

### *Tlx* is absent from available genomes of cnidarians lacking a medusa. Reduced medusae are often referred to as eumedusoids, cryptomedusoids, or sporosacs[26] depending on their degree of developmental arrest. Here we code reduced medusae as eumedusoids if they exhibit a gastrovascular system, velum and tentacles but lack discrete gonads and a mouth, cryptomedusoids if they bear only radial canals and highly reduced tentacle processes and sporosacs if they represent a fixed gonophore lacking any medusa features. Some hydrozoans, such as *Hydra*, do not bear any gonophores and instead release their gametes directly from the body column of the polyp. Given that eumedusoids possess many medusa-specific characteristics, we consider, for the purpose of this study, those species bearing eumedusoids as having a medusa stage, whereas those bearing cryptomedusoids, sporosacs, or absence of any gonophore, we consider lacking a medusa stage. For siphonophores, there is no counterpart to a medusa, rather medusa-like structures are found in the nectophores and the gametes are in the gonophore. However, given the presence of medusa-like structures in the nectophores, we coded them as eumedusoids.

In our search for the *Tlx* gene in publicly available cnidarian draft genome assemblies from 71 species, we found that the presence of *Tlx* is invariably correlated with the presence of a medusa in the cnidarian life cycle. Specifically, *Tlx* was found in

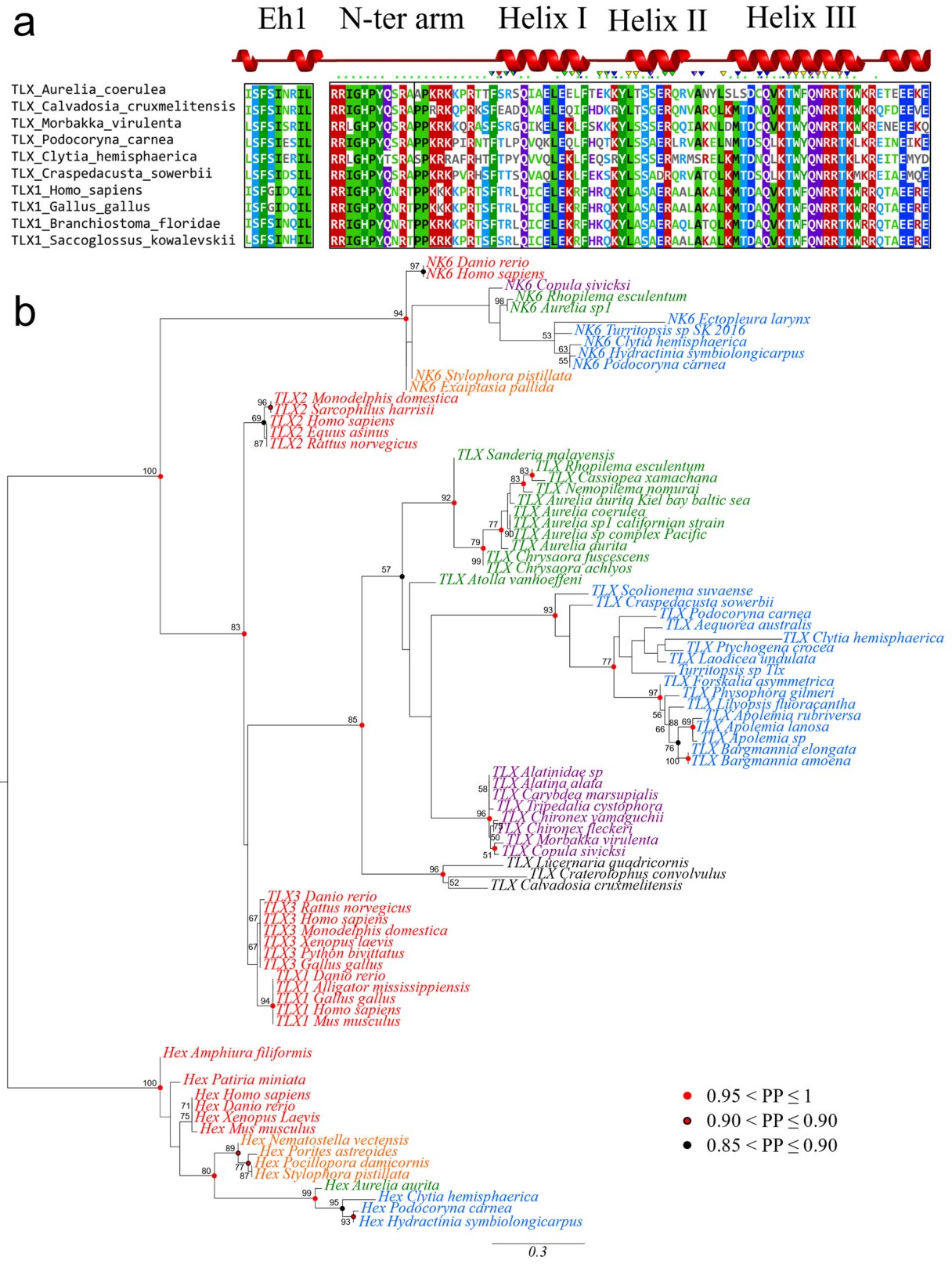

all 28 of the draft genomes from species that have medusae and not found in any of the 43 available draft genomes from species that lack medusae, including all anthozoans and endocnidozoans and the six hydrozoans that have lost the medusae stage (Table 1).

*Tlx* was also searched for in the transcriptomes from 186 cnidarian species, wherein the gene was present in 48 of 73 transcriptomes from medusa-bearing species (Table 1). The

failure to recover *Tlx* in some of the transcriptomes of medusa-bearing species is likely the result of the particular tissue and/or developmental stage from which the transcriptome was generated and/or insufficient depth of sampling.

*Tlx* was not present in any of the available transcriptomes from species that lack a medusa, with three exceptions (*Millepora squarrosa*, *Ectopleura larynx* and *Dynamena pumila*) (Table 1).

**Fig. 1 TLX domains conservation and orthology relationship to bilaterian TLX. a** Schematic of TLX protein structure and the corresponding amino acids sequence alignment for TLX conserved domains from six representative medusozoans and four bilaterians. The protein structure annotation was performed using SAS[65], using *Podocoryna carnea* TLX sequence. Helices represent predicted helix secondary structures; green squares represent residues predicted to interact with DNA and the triangle represents predicted active sites. Highly conserved positions are highlighted (>70% identity) in the alignment. Colors in the alignment represent features of the position, purple (polar uncharged), red (positively charged), blue (negatively charged) and green (hydrophobic). **b** Phylogram from maximum likelihood analysis, with three NK-L representatives (*Tlx, Nk6* and *Hex*). Vertebrate sequences are in red, hydrozoan sequences in blue, staurozoan sequences in black, scyphozoan sequences in green, cubozoan sequences in purple and anthozoan sequences in orange. Bootstrap values greater than 50% are indicated (1000 bootstraps) next to the nodes. Bayesian posterior probabilities greater than or equal to 85% are reported on the nodes with colored circles (color code on the figure). *Hex* sequences are used as the outgroup. Scale bar = number of inferred substitutions per position in the alignment.

**Table 1 Tlx presence in publicly available cnidarian genomes and transcriptomes.**

|  | Tlx in genomes | Tlx in transcriptomes |
|---|---|---|
| Anthozoa | 0 out of 31 | 0 out of 101 |
| Hexacorallia | 0 out of 29 | 0 out of 75 |
| Octocorallia | 0 out of 2 | 0 out of 20 |
| Cerianthария | N/A | 0 out of 6 |
| Endocnidozoa | 0 out of 6 | 0 out of 4 |
| Hydroidolina without medusae[a] | 0 out of 6 | 3 out of 12 |
| 'Anthoathecata' | 0 out of 6 | 2 out of 11 |
| Leptothecata | N/A | 1 out of 1 |
| Hydroidolina with medusae | 7 out of 7 | 30 out of 49 |
| Siphonophorae | N/A | 14 out of 32 |
| Leptothecata | 1 out of 1 | 10 out of 10 |
| 'Anthoathecata' | 6 out of 6 | 6 out of 7 |
| Trachylinae | 2 out of 2 | 1 out of 4 |
| Trachymedusae | N/A | 0 out of 2 |
| Narcomedusae | N/A | 0 out of 1 |
| Limnomedusae | 2 out of 2 | 1 out of 1 |
| Scyphozoa | 12 out of 12 | 8 out of 10 |
| Discomedusa | 12 out of 12 | 7 out of 8 |
| Coronata | N/A | 1 out of 2 |
| Staurozoa | 3 out of 3 | 4 out of 5 |
| Cubozoa | 4 out of 4 | 5 out of 5 |

[a]Medusae defined as any medusa-like structure (nectophore, eumedusoid, medusa) present in the life cycle of the species.

*Millepora squarrosa* (Anthoathecata, Milleporidae) *Tlx* is likely a pseudogene as it exhibits premature stop codons and the *Dynamena pumila* (Leptothecata, Sertulariidae) sequence was a partial sequence and thus the presence of a complete coding sequence is unknown. Although *Ectopleura larynx* (Anthoathecata, Tubulariidae) has the characteristic *Tlx* domains, it exhibits a unique codon insertion, leading to an asparagine in the highly positively charged N-terminus of the homeodomain. Whether this change could affect *E. larynx Tlx* function is unknown.

Due to the sampling restrictions imposed by screening *Tlx* from publicly available cnidarian genomes, and the limitations of transcriptomes for estimating the presence of a gene as discussed above, we used degenerate PCR to screen genomic DNA from 100 medusozoan taxa for the presence of *Tlx*, including all medusozoan suborders, in order to span the breadth of medusozoan diversity. This PCR genomic screening for the *Tlx* sequence was entirely used to code for the presence and absence of the gene as displayed in Fig. 2. The primers were designed to amplify the N-terminal arm and the entire homeobox region. In the 69 taxa surveyed that have a medusa (or eumedusoid), a *Tlx* gene fragment was successfully amplified in 59 (86%). The failure to amplify a *Tlx* fragment in the other 10 medusa-bearing taxa could be due to the limitations of degenerate PCR, which is highly

sensitive to DNA quality and primer binding. An amplification product was not obtained in 28 out of 31 taxa (90%) that lack a medusa (sporosac, cryptomedusoid or no gonophore). The three non-medusa bearing species for which a *Tlx* fragment was recovered were the cryptomedusoid bearing *Ectopleura larynx*, also found in the transcriptome above, as well as two sertulariid species that bear sporosacs (*Amphisbetia minima* and *Sertularia perpusilla*). The sequence from *Sertularia perpusilla* (Leptothecata, Sertulariidae), like the sequence of *Millepora squarrosa* discussed above, is likely a pseudogene as it contains several premature stop codons. Thus, of the total of five TLX sequences isolated from non-medusae bearing species from transcriptomes and/or PCR, only *Amphisbetia minima* (Leptothecata, Sertulariidae) has a typical TLX sequence within the amplified region.

**The absence of the *Tlx* gene is correlated with the absence of the medusa in Hydrozoa.** Using the medusozoan phylogenetic tree from Cartwright and Nawrocki[27] we pruned the taxa to match those 100 samples for which degenerate PCR data were generated and reconstructed the evolution of the medusa stage. Similar to findings of Cartwright and Nawrocki[27] our analysis inferred several independent instances of reduction and loss of the medusa stage. Out of the 17 such cases, nine are reductions to sporosacs, five to eumedusoid, one to cryptomedusoid and two are complete losses of the gonophore (Fig. 2).

A Bayesian correlation analysis of the presence of *Tlx* and the presence of medusa-like structures (medusa and eumedusoid) shows a very strong correlation between the two traits (Log Bayes factor = 12.645926). The same analysis also supports the presence of TLX together with the medusa stage as ancestral in Medusozoa (PP = 1) (Fig. 2).

**The TLX homeodomain shows evidence of relaxed selection in species lacking medusa-like structures.** To further investigate the apparent conservation of the TLX homeodomain amongst medusa bearing and the few non-medusa bearing lineages, we tested for relaxation/intensification of selection on the *Tlx* homeodomain in a codon-based phylogenetic framework. Using selection analyses, we tested the four *Tlx* sequences that were found from non-medusa-bearing species (*Millepora squarrosa, Ectopleura larynx, Dynamena pumila, Amphisbetia minima*) for relaxed selection. The *S. perpusilla* sequence was removed from the analysis as the aberrant sequence did not allow for proper codon alignment. When testing these four sequences against the 46 medusa bearing reference species sequences, we found strong evidence for a relaxation of selection on *Tlx* in those medusa-less lineages ($K = 0.15$. $p = 0.0000$). While testing those four lineages independently, the same evidence of relaxation of selection was found except for *E. larynx* ($K = 1.19$, $p = 0.1715$) for which a non-significant intensification was inferred. By contrast, a significant intensification of selection on the *Tlx* homeodomain was detected for Acraspeda (Scyphozoa, Cubozoa and Staurozoa) ($K = 1.66$, $p = 0.0000$) and the hydrozoan order Siphonophorae

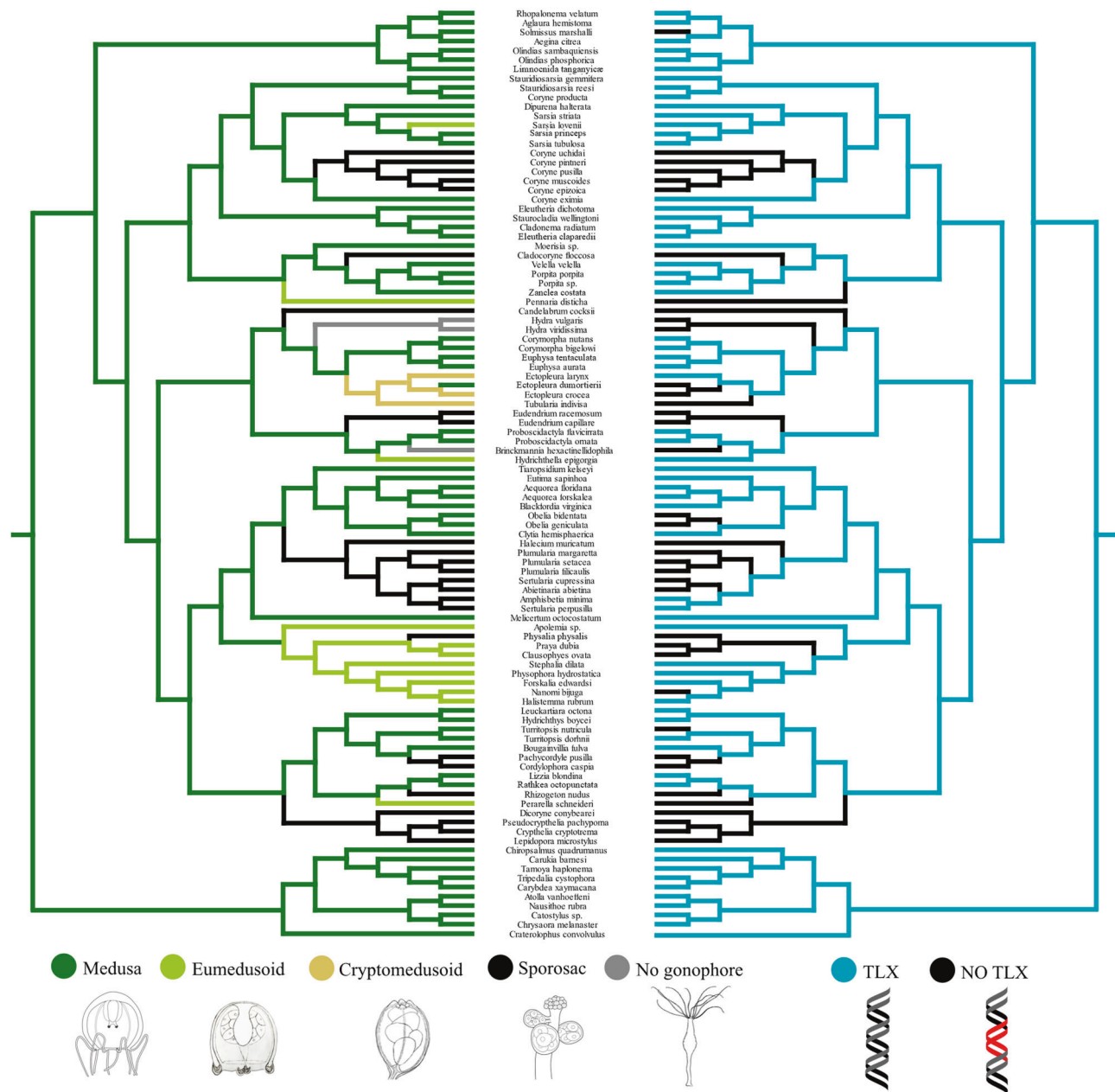

**Fig. 2 Wagner maximum parsimony ancestral character state reconstructions of medusozoan reproductive systems (left) against the Dollo maximum parsimony ancestral character state reconstructions of the presence of *Tlx* (right).** Characters are unweighted and unordered, no optimization model was applied. Branches are colored in terms of degree of medusa reduction and presence of *Tlx* (shown in legend). Phylogeny pruned from ref. [27] and rooted with Acraspeda.

($K = 2.30$, $p = 0.0050$), using medusa-bearing hydrozoan species as a reference. No significant trend in selection was detected for the other hydrozoan lineages (see Table 2). This relaxation of selection in Hydrozoa may in part explain the pattern of multiple medusa losses that is not found in the other medusozoan classes.

Lastly a FUBAR test (Fast, Unconstrained Bayesian Approximation[28]) was performed to identify site-specific variation in the selection of cnidarian TLX. Unsurprisingly, 9 codons in the EH1 domain and 75 codons in the homeodomain (including the N-terminal arm and flanking regions), respectively, showed evidence for pervasive purifying selection (PP > 0.99), and no phylum-wide diversifying selection was detected upon remaining sites. Interestingly, the analysis detected smaller motifs flanking the homeodomain and its N-terminal arm. Among scyphozoans, a significant (PP > 0.99) episodic purifying selection

was detected on sites flanking the homeodomain (PWQILXK upstream of the N-terminal arm and TEEEKEEQRHAL downstream of the homeodomain), while a significant (PP > 0.99) episodic purifying selection was detected in the flanking regions of the homeodomain of hydrozoan TLX, corresponding to a highly conserved CXC motif upstream of the N-terminal arm and a EINEMXEQQXR motif downstream of the homeodomain. Flanking motifs found in hydrozoans and scyphozoans do not provide direct information regarding the binding target of the homeodomain but could suggest that the target or the affinity for the target of TLX might differ between these lineages.

**Expression of *Tlx* is upregulated during medusa development.** To investigate the expression profile of *Tlx* in the medusozoan life

**Table 2 Analyses of the intensity of the selection on the TLX homeodomain within medusozoans.**

| Reference branches<br>Test branches | Medusozoans with medusae | | Hydrozoans with medusae |
|---|---|---|---|
| Hydrozoans without medusae | $K = 0.15$, $p = 0.0000$ | Acraspeda | $K = 1.16$, $p = 0.0000$ |
| *Amphisbetia minima* | $K = 0.00$, $p = 0.0010$ | Trachylina | $K = 0.78$, $p = 0.413$ |
| *Dynamena pumila* | $K = 0.16$, $p = 0.0000$ | Siphonophorae | $K = 2.30$, $p = 0.0050$ |
| *Ectopleura larynx* | $K = 1.19$, $p = 0.7710$ | Leptothecata | $K = 1.26$, $p = 0.521$ |
| *Millepora squarrosa* | $K = 0.66$, $p = 0.0249$ | Anthoathecata | $K = 0.72$, $p = 0.195$ |
| Amplicons from degenerate PCR | $K = 0.71$, $p = 0.1715$ | Amplicons from degenerate PCR | $K = 1.04$, $p = 0.917$ |

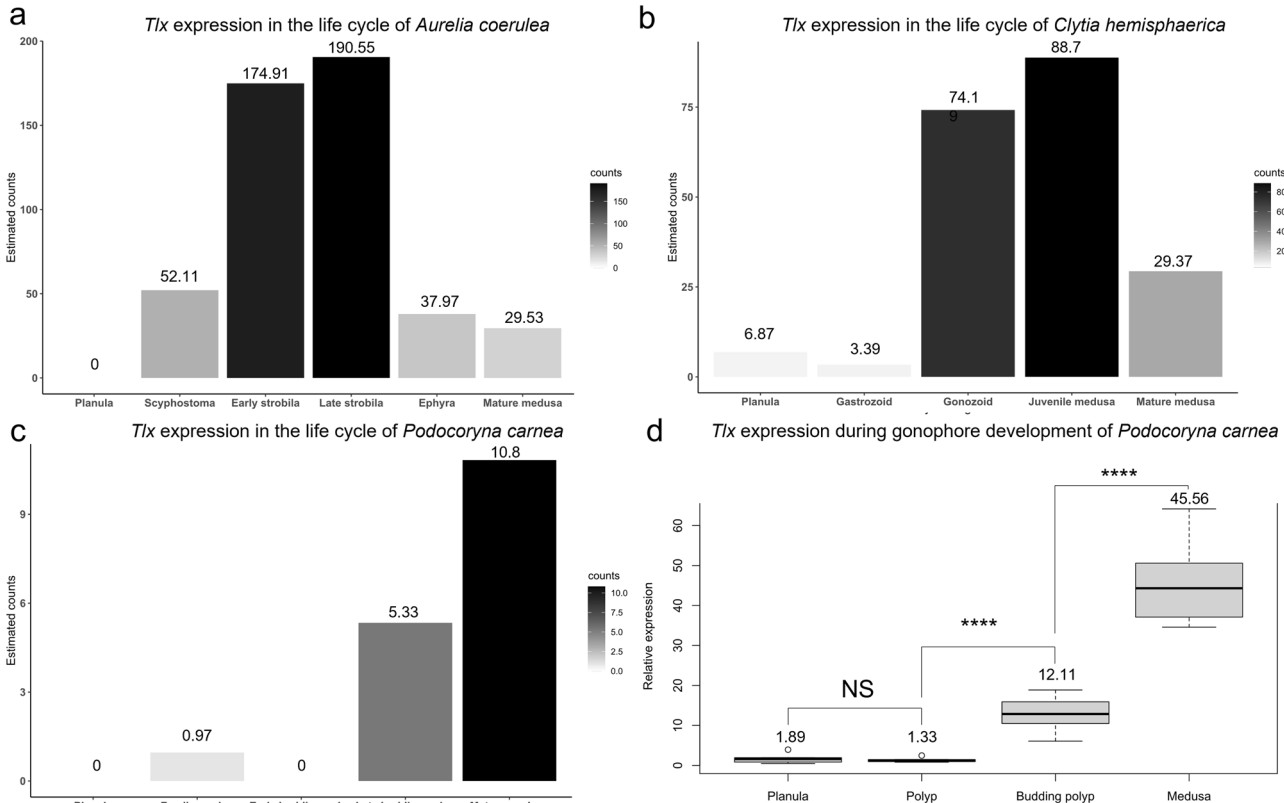

**Fig. 3 Differential expression of *Tlx* in the life cyle of three distantly related medusozoans: *Aurelia coerulea*, *Clytia hemisphaerica* and *Podocoryna carnea*.** Normalized expression of *Tlx* in corrected counts for life cycle developmental stages (**a–c**). **a** *Aurelia coerulea*, **b** *Clytia hemisphaerica*, **c** *Podocoryna carnea*. The gradient charts indicate the breadth of *Tlx* corrected counts values for each species. The differential expression patterns of *Tlx* are all supported by maximum posterior probabilities (FDR = 0.0). **d** RT-qPCR in *Podocoryna carnea*. (****) indicates a two tailed *p*-value <0.0001 from t-tests, feeding polyp-Late budding polyp (*p*-value = 1.23e−08), late budding polyp-medusa (1.501e−08), planula-polyp (NS) indicates a non-significant two tailed *p*-value (*p*-value = 0.2995) from unpaired t-test. Planula (*n* = 7, independent cDNA samples), feeding polyp (*n* = 8, independent cDNA samples), budding polyps ((*n* = 8, independent cDNA samples), medusa ((*n* = 8, independent cDNA samples).

cycle, we performed differential expression (DE) analyses on distinct life cycle stages for the scyphozoan *Aurelia coerulea*, and two medusae-bearing hydrozoan species, *Podocoryna carnea* and *Clytia hemisphaerica*. For *Aurelia*, *Tlx* expression is first detected in the polyp (scyphostoma), peaks when the polyp is producing medusae (strobilating) and is then downregulated in the juvenile medusa (ephyra) and mature medusa stages to expression levels comparable to the scyphostoma (PP = 1, Fig. 3a). In *Clytia hemisphaerica*, *Tlx* expression is first detected at low levels during the planula stage and is maintained at low level in the feeding polyp (gastrozooid). *Tlx* expression is upregulated in the reproductive polyp that buds medusae (gonozooid) and the newly released juvenile medusa. In the adult medusa of *Clytia*, the expression of *Tlx* is downregulated to an intermediate level (PP = 1, Fig. 3b). In *Podocoryna carnea*, no significant expression

of Pc*Tlx* is detected at the planula stage, in the non-reproductive feeding polyp nor in the reproductive polyp when the medusae buds are initially detected. Pc*Tlx* expression is upregulated in reproductive polyps budding medusae, during later stages of medusae development and remains at this expression level in the fully developed medusa after it is released from the polyp (PP = 1, Fig. 3c). Although DE patterns were significant for the three species using estimated counts, other metrics, namely Transcripts per million (TPM) and Fragments Per Kilobase Million (FPKM), were inconclusive, likely due to the overall low expression of *Tlx*. To validate the RNA-Seq results in *P. carnea*, we performed RT-qPCR on planulae, non-reproductive polyps, budding polyps and released medusae of *P. carnea* and found a significant difference in the relative expression of *Tlx* between the four life cycle stages (ANOVA, *p* < 0.0001), as well as a higher expression of *Tlx* in

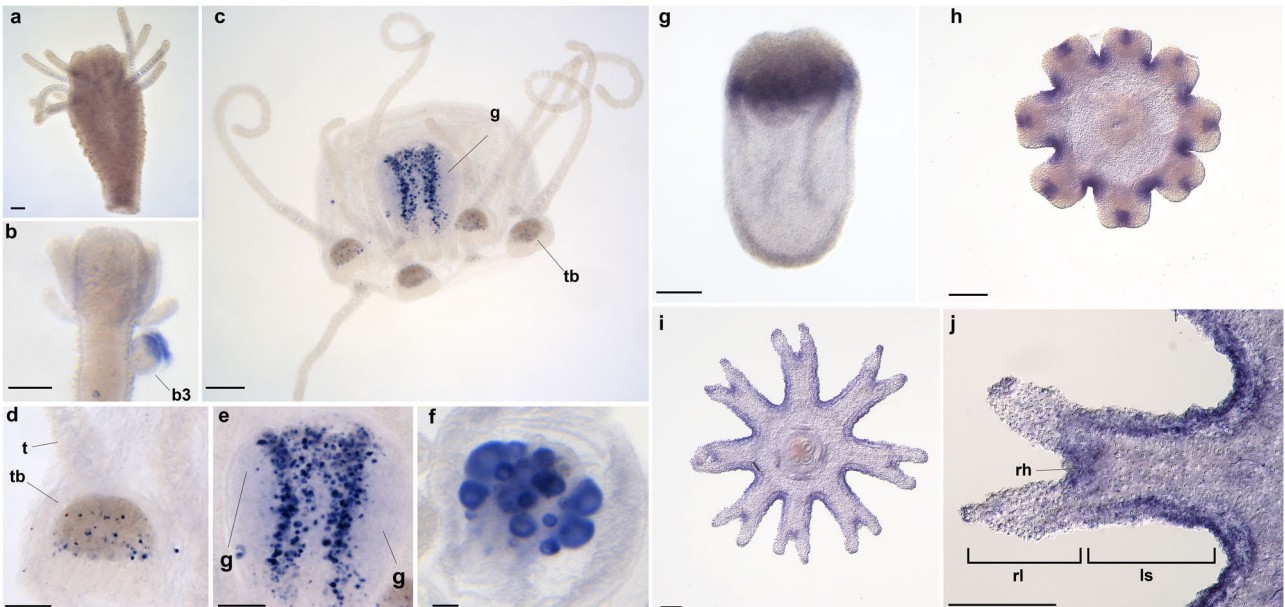

**Fig. 4 Tlx expression pattern in the hydrozoan Podocoryna carnea and the scyphozoan Pelagia noctiluca.** In situ localization of *PcTlx* transcripts (purple-blue stain) in *Podocoryna carnea* (**a–f**) and *PnTlx* transcripts in *Pelagia noctiluca* (**g–j**). **a** *P. carnea* male non-reproductive polyp, **b** *P. carnea* male budding polyp, **c** *P. carnea* male medusa, **d** higher magnification of **c** of a tentacle bulb, **e** higher magnification of **c** of the manubrium, **f** oocytes in a *P. carnea* female medusa. The manubrium (**d**) is presented oral side down. The tentacle bulb (**d**) is presented proximal side down. **g** *P. noctiluca* planula larva, **h** *P. noctiluca* pre-ephyra, **i** *P. noctiluca* young ephyra and **j** *P. noctiluca* young ephyra marginal lappet. The planula is presented side oral up. **h–j** are presented in oral view. Abbreviations: b3 medusa bud stage 3, g gonad, ls lappet stem, rh rhopalia, rl rhopalial lappet, t tentacle, tb tentacle bulb. Scale bar: 200 μm (**a–c**), 100 μm (**d–j**).

released medusae compared to budding polyps (t-test, $p < 0.0001$), (see Fig. 3d).

**Tlx expression is spatially restricted during medusa development.** *Tlx* expression patterns were first determined by whole mount in situ hybridization in the hydrozoan *Podocoryna carnea* and the scyphozoan *Pelagia noctiluca*. *P.carnea* medusa buds were staged according to Frey[29] with stage 1 being the earliest detection of a bud to stage 6 as the latest stage, right before release from the polyp. Expression in *P. carnea* was detected in medusa buds and free-living medusae (Fig. 4b–f), but not in polyps (Fig. 4a). *Tlx* was detected in the distal portion of the medusa bud (stages 3-4) in both the endoderm and ectoderm as well as the cellular membrane surrounding the bud (Fig. 4b). *PcTlx* was not detected in early buds (stages 1-2) or in late buds (stages 5-6). *Tlx* expression was again detected in free-living medusae in an endodermal cell subpopulation surrounding the developing gonads (Fig. 4c, e) and additionally in the oocytes of female medusae (Fig. 4f), although oocyte staining was also detected in the control (sense probe) (Supplementary Fig. S5d). *Tlx* was sporadically detected in isolated ectodermal cells in the manubrium as well. In older medusae, the expression was also detected in the endoderm of all of the tentacle bulbs (structure on the bell margin proximal to the tentacles) (Fig. 4d).

In *P. noctiluca*, the planula larva develops directly into a juvenile medusa (ephyra) with the oral most portion of the planula giving rise to the subumbrella while the sub-oral portion of the planula develops into the bell margin anlage and associated structures. The ephyra possesses lappets, which are marginal extensions of central disk. The distal portion of the lappets bears a sense organ called a rhopalium (see ref.[30] for the description of *P.noctiluca*'s anatomy).*Tlx* is first detected in the planula in a suboral ring-like pattern (Fig. 4g). In the pre-ephyra, *Tlx* expression is restricted to the pits of the marginal lappets and developing rhopalia (Fig. 4h). In the young ephyra *Tlx* expression

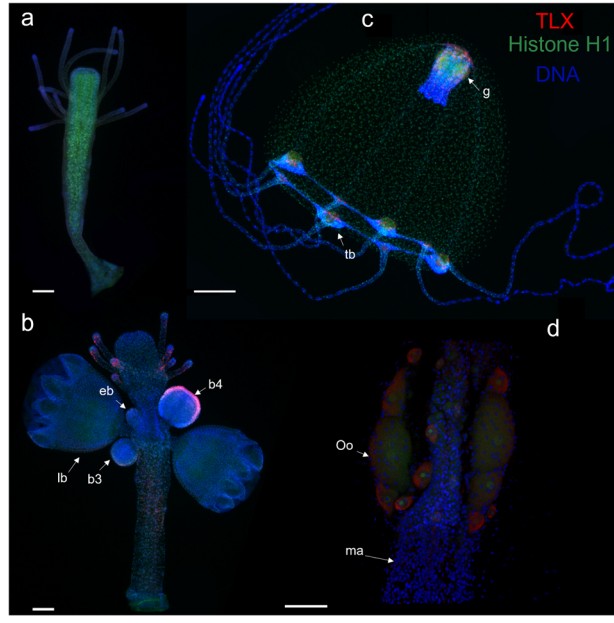

**Fig. 5 Localization of the TLX protein in Podocoryna carnea.** Immunostaining of TLX (red) and histone H1 (green) during the medusa development in *Podocoryna carnea* (**a–d**). **a** Non-reproductive, **b** released male medusa, **c** budding polyp, **d** manubrium of a mature female medusa. All samples are counter stained for DNA with Hoechst (blue). Abbreviations: b3 medusa bud stage 3, b4 medusa bud stage 4, g gonad, lb late bud (stage 5 and above), ma manubrium, oo oocyte, tb tentacle bulb. Scale bar: 200 μm (**a–c**), 50 μm (**d**).

is detected in the lappet stems (Fig. 4j, i) and in the proximal portion of the rhopalium (Fig. 4j).

To detect TLX protein during medusa development in *Podocoryna carnea*, we used polyclonal antiserum generated

against a portion of PcTLX. The non-reproductive polyps, early medusae (stage 1), and later budding stages (5 and above) do not show any TLX expression (Fig. 5a, b). TLX expression is observed in the distal portion of medusa of stage 4 buds (Fig. 5b), consistent with in situ hybridization patterns. In reproductive polyps TLX was detected in the tentacles of the polyp (Fig. 5b). In released medusae, TLX is localized to the tentacle bulbs and manubrium, consistent with observed in situ patterns (Fig. 5c). In tentacle bulbs, TLX has a scattered expression pattern although more concentrated to the proximal portion of the bulb (Fig. 5c, Supplementary Fig. S4c). In the manubrium, TLX is expressed in a scattered pattern throughout the manubrium (Fig. 5b). TLX positive cells were sporadically observed in the lumen of radial and circular canals (Supplementary Fig. S4b). In female medusae, TLX was detected in the oocytes (Fig. 5c, d). In all TLX-positive cells, the protein was localized to the cytoplasm, which is unexpected for a transcription factor (Fig. 5d and Supplementary Fig. S4b). The cytosolic location of TLX in our immunostaining could be due to a transient nature of nuclearization of TLX or inaccessibility of the epitopes in the nucleus.

## Discussion

Our detailed analysis of phylogenetic distributions of the medusa stage and the *Tlx* sequence reveals a striking correlation between the presence of an intact *Tlx* gene and the presence of the medusa life cycle stage in cnidarians. The few occurrences of *Tlx* in non-medusa-bearing species are characterized by non-sense mutations and/or relaxed selection in conserved regions, suggesting conversion to pseudogenes and possible loss of TLX function in those species. Interestingly, Acraspeda and Siphonophorae both appear to have retained medusa-like structures in their life cycle throughout their evolutionary history with very few exceptions (see refs. [31–33]). This overall maintenance of medusa-like structures in these two lineages might explain the intensification of the selection upon the TLX homeodomain.

The conservation of the structure of TLX between cnidarians and bilaterians can shed light on the ancestral role in signaling of TLX. The EH1 domain has been shown to interact with the corepressor Groucho/TLE and act as a transcriptional repressor in bilaterians[34,35]. The conservation of the EH1 domain of TLX in cnidarians suggests a conservation of that role in transcriptional repression. The N-terminal arm has been shown to interact with ARX in bilaterians[36,37]. Its conservation suggests that this is also the case in medusozoans. Together, these results suggest that TLX possessed an EH1 domain and N-terminal arm in the last common ancestor of cnidarians and bilaterians. Putatively, TLX was ancestrally functioning in transcriptional repression together with ARX.

In our phylogenetic analyses, we recovered a *Tlx-like* orthology group, that lacks the N-terminal arm and EH1 domain, within a well-support clade containing the cnidarian and bilaterian *Tlx* genes (Supplementary Fig. S2). Given that some cnidarian taxa possess both a *Tlx-like* and *Tlx* gene and that the *Tlx-like* gene is found in sponges, *Tlx* likely emerged following a duplication event from the *Tlx-like* gene. The relative position of ctenophores and sponges remains uncertain, thus the absence of the *Tlx-like* gene in ctenophores could either suggest that the *Tlx-like* gene emerged in the last common ancestor of Metazoa and was lost in ctenophores or emerged in the last common ancestor of a putative clade composed of Porifera, Cnidaria and Bilateria (Fig. 6).

Our RNA-seq analyses show striking upregulation of *Tlx* during medusa development in three disparate medusozoans, suggesting the existence of a conserved role of *Tlx* in the development of the medusa, despite the very distinct developmental trajectories of these three species. Specifically, while the hydrozoan medusae relies on lateral budding from the polyp, *Clytia* possesses dedicated polyps

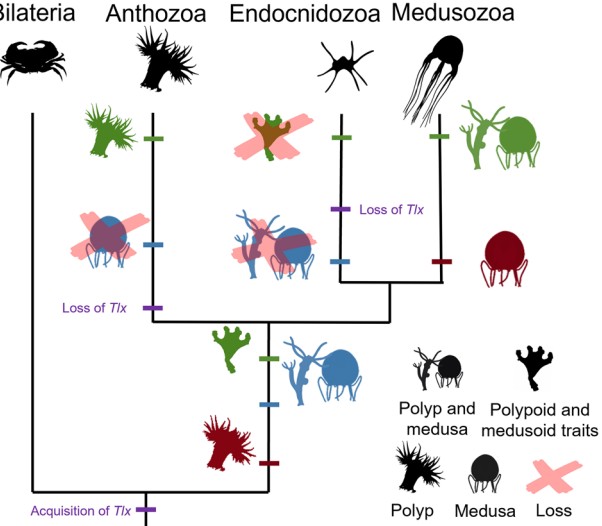

**Fig. 6 Summary of the hypotheses for the evolution of the medusa stage in conjunction with *Tlx*.** Metagenetic life cycle with a polyp and medusa stage ancestral for Cnidaria (blue). The medusa stage as a derived traits of Medusozoa from an ancestral monogenetic life cycle possessing a polyp stage (red). Ancestral monogenetic cnidarian possessing polypoid and medusoid traits (green).

(gonozooids) for budding medusae, whereas *Podocoryna* transforms its feeding polyp to a reproductive polyp upon onset of medusa budding. *Aurelia* does not bud medusae but undergoes a process of transverse fission of its polyp, called polydisc strobilation.

Spatial expression patterns between two distantly related medusozoans highlight a dual role of *Tlx* in patterning the early medusa and later in areas of active development/cellular differentiation. In *Podocoryna carnea*, PcTlx is detected ephemerally in the distal portion of the developing bud (stage 3-4) where most of the morphogenesis takes place[29]. This pattern of expression is reminiscent in timing and location to the onset of the *Wnt* organizer[14], although in a broader expression domain. This suggests that *Tlx* may have a role in the general patterning of the oral-aboral axis of the bud where the identity of the oral structures of the medusa is established. *Pelagia noctiluca* lacks a polyp life cycle stage, instead the planula larva develops directly to a pre-ephyra (juvenile jellyfish). The early expression in the planula larva is consistent with a role of *Tlx* involved in the patterning of medusae structures that takes place in the oral region of the planula of *P. noctiluca*.

Although axial patterning of newly released hydromedusae and scyphozoan ephyrae is mostly complete, additional development of tentacles, sense organs and nematocytes (stinging cells) continue. In hydromedusae, the tentacle bulbs serve as regions of active cell proliferation, neurogenesis and nematogenesis (see refs. [38–40] and Supplementary Fig. S4d). In *Podocoryna*, TLX is expressed in those territories and most distinctively in rows of cells proximal to the connection of the radial canal to the bulb (see Supplementary Fig. S4c) suggesting TLX could be involved in the aforementioned processes.

In *P. noctiluca* ephyrae *Tlx* is expressed broadly in territories undergoing patterning, notably the developing rhopalia. The expression of *Tlx* in the proximal portion of the rhopalar canal suggests a similar role in cell differentiation as in *P. carnea*. Akin to the tentacle bulb of hydromedusae, the proximal-most portion of the rhopalium is characterized by a dense nervous system[41] and active cell proliferation. Although no expression patterns were documented in reproductive medusa of *Pelagia* due to its large size, *Tlx* expression in the reproductive medusa of

*Podocoryna* was found to be localized to the manubrium, both in cell populations distinct from the germline as well as in mature oocytes in the medusae. Together these results suggest that *Tlx* plays a role in the early patterning of the developing medusa, then in cellular differentiation, and an additional role in oocyte maturation or as a maternal effect gene.

The phylogenetic distribution and expression pattern suggests that *Tlx* is intimately tied to medusa development. Given that the presence of *Tlx* appears to be ubiquitous in bilaterians, *Tlx* was likely present in the last common ancestor of Bilateria and Cnidaria and then secondarily lost multiple times in cnidarians, including at the base of Anthozoa, Endocnidozoa and independently several times in Hydrozoa. The ancestral presence of the *Tlx* gene in Cnidaria, in conjunction with the striking correlation of the gene *Tlx* with the presence of the medusa stage in medusozoans, and its apparent role in medusa development, suggests several possible scenarios regarding the ancestral life cycle in Cnidaria. *Tlx* and a metagenetic life cycle exhibiting a polyp and medusa stage could have been present in the ancestor of Cnidaria, with the medusa lost multiple times in cnidarian evolution, including at the base of Anthozoa, Endocnidozoa and multiple times in Hydrozoa (Fig. 6 blue). Fossil records from the early Cambrian exhibiting discrete medusa and polyp stages could support the ancestral presence of such a life cycle[42,43]. An alternative explanation is that *Tlx* could have had an unknown ancestral function in the last common ancestor of Cnidaria and Bilateria and was rapidly exapted in medusozoans for medusa development and/or medusa-specific structures or cell types (Fig. 6 red). In this scenario, the ancestral function is no longer necessary in anthozoans and other cnidarians that lack *Tlx*. A third explanation is that *Tlx* had an ancestral function in the development of specific structures in a cnidarian monogenetic ancestor that exhibited both polypoid and medusoid features. In that scenario, the emergence of a discrete polyp stage in anthozoans, reduction of the ancestral body plan in parasitic endocnidozoans, and the uncoupling of these features into two discrete generations in medusozoan could have been responsible for the pattern of loss and maintenance of *Tlx* respectively (Fig. 6 green). The presence of the late Ediacarian fossil *Haootia quadriformis*, that exhibits both polypoid and medusoid traits would be consistent with this scenario[44]. Further investigations of *Tlx* function in medusozoans and in early diverging bilaterians may help clarify the ancestral role of *Tlx* in Cnidaria.

## Methods

**Phylogenetic methods**. Amino acid sequence alignments were carried out with Muscle[45] using default parameters and manually refined on MEGA7[46]. The EH1 domain, N-terminal arm and homeodomain of TLX were aligned with the homeodomain of NK-L members NK6 and HEX, totaling 82 taxa and 83 characters. Phylogenetic analyses were conducted respectively under MrBayes 3.2.7[47] and RaxML[48], under the model LG + I + G with 4 discrete gamma categories as selected by Modeltest-NG (based on BIC), on CIPRES portal[49]. For the Bayesian phylogenetic analysis, 4 runs and 6 Markov chains were generated, and the analysis was run for 5 million generations with a 25% burn-in. The posterior probabilities, as well as the final topology come from a majority consensus of the sampled trees. For the Maximum likelihood phylogenetic analysis, support values were evaluated by non-parametric bootstraps (1000 replicates). Bootstrap support values were reported when above 50 and posterior probabilities ranging from 0.90 to 1 are indicated at the node in respect to the color coding (Fig. 1b). The second and third best hits resulting from the reciprocal blast during gene candidate searches were selected as outgroups.

**Ancestral character state reconstruction of *Tlx* and the gonophore in Medusozoa**. The MP ancestral character state reconstruction of the presence of *Tlx* in genomic samples and the gonophore were conducted under Mesquite v3.61[50] on a pruned tree from ref. [27], re-rooted with Acraspeda. Dollo parsimony was imposed for *Tlx* ancestral character reconstuction as convergent evolution or horizontal gene transfer seemed unlikely. The reconstruction of the gonophore ancestral state was conducted under Wagner parsimony; characters were unweighted and unordered. Character states were coded as such: no gonophore, sporosacs, cryptomedusoids, eumedusoids, and medusa. In *Ectopleura larynx*, *Tlx* could not be

amplified by degenerate PCR but was found in publicly available transcriptomic data, and thus was coded as present (only one case, *Ectopleura larynx*). Within siphonophores three different structures have been proposed to be homologous to the hydromedusa. The eudoxid is a sexual free-living individual, the nectophore is an asexual swimming zooid exhibiting a velum, radial and circular canals, and the gonophore is the sexual zooid that may or not exhibit medusa-like structures depending on lineages (see ref. [51]). Here the presence of either nectophores or eudoxids was coded as eumedusoid. Given the peculiar uncoupling of medusa-like structures (such as seen in nectophores) and the germline (gonophores) into two different zooid types, the homology of these structures to the gonophore of other hydrozoans remains unclear. Thus, the developmental mechanisms of these zooid types, especially in the context of heterochrony and the role of *Tlx*, might differ from other hydrozoans.

**Cnidarian genomes and transcriptomes assembly**. Most of the genome and transcriptome sequences were obtained from the NCBI sequence read archive (SRA) (Supplementary Data 1) and required assembly in order to screen for *Tlx*. Each library was trimmed of low-quality reads and adapters using fastp[52]. For those transcriptomes from different libraries, filtered reads were combined into a single dataset followed by de novo transcriptome assembly using Trinity v2.8.5[53]. Genome assemblies were carried out using Spades v3.13.1[54]. Genomes that did not require assembly were obtained from NCBI. The source of all the genomes and transcriptomes used in this study can be found in Supplementary Data 1.

**In silico search for *Tlx***. Potential orthologs of *Tlx* were identified through reciprocal blasts of TLX amino acid sequences (tblastn, e-value cut-off set to $10^{-80}$ and $10^{-10}$) from *Chironex fleckeri, Calvadosia cruxmelitensis, Aurelia aurita, Clytia hemisphaerica, Podocoryna carnea, Agalma elegans,* and *Craspedacusta sowerbii* against cnidarian transcriptomes and genomes.

**Degenerate PCR screening for *Tlx* in cnidarian genomic DNAs**. Degenerate primers in the second exon of *Tlx* (Forward 5'-GGNCAYCCNTAYCAVAGC/ MGNGC-3', Reverse 5'-GTKCKHCKRTTYTGAWACCA-3') were designed from eight medusozoan species: *Chironex fleckeri, Calvadosia cruxmelitensis Aurelia aurita, Clytia hemisphaerica, Podocoryna carnea, Agalma elegans* and *Craspedacusta sowerbii*. The primers span the N-terminal arm to the C-terminal end of TLX homeodomain, for a total expected amplicon length of 196 bp. Degenerate PCR were carried out using OneTaq 2X Master Mix Standard Buffer, according to manufacturer instructions and with an annealing temperature of 40 ℃. Amplification products were electrophoresed in a 1.2% agarose gel to assess the presence of the amplicon. The amplifiability of the genomic samples was assessed using 16S degenerate primers. Ten amplicons were selected at random, cloned into the pCR4-TOPO plasmid and sequenced to confirm TLX identity.

**Bayesian correlation analysis of the presence of *Tlx* and the medusa stage**. Character coding can be found in Supplementary Data 1. Correlation analyses between the presence of *Tlx* and medusa stage was performed using BayesTraits v3[55], imposing irreversibility for *Tlx* by setting transition rate for regains of *Tlx* to zero ($q_{13} = 0$, $q_{24} = 0$). The presence of *Tlx* and the medusa stage were coded as binary characters. The presence of sporosacs and cryptomedusoids as well as no gonophores were coded as absent and the eumedusoids and fully developed medusae were coded as present. Although eumedusoids do not feed, they have nearly all other medusa components and thus were treated as medusae. The statistical support for the correlation analysis was carried out by computing the marginal likelihood of the two alternative models, independence and dependence of *Tlx* with the medusa stage. The calculation of the Log Bayes Factor was performed and interpreted as recommended by the BayesTraits manual. According to the Bayestraits manual logBF can be interpreted as such: logBF <2 Weak evidence, logBF >2 Positive evidence, 5 ≤ logBF ≤ 10 Strong evidence, logBF >10 Very strong evidence.

**Animal care**. *P. carnea* colonies were grown on microscope slides contained in slide racks and kept in artificial seawater (REEF CRYSTALS, Aquarium Systems) in a 7 L Kreisel tank at room temperature (~18 °C) with a salinity of 29 ppt. Male and female colonies were kept in separate species. Colonies were fed two-day old *Artemia* nauplii twice a week and blended mussels once a week. Unfed one and three day old released medusae were collected. Prior to every experiment, *P. carnea* colonies were starved for four days. Animals were relaxed for 30 min by addition of menthol crystals (1 mg/ml) to the medium and fixed after two medium changes.

**Probe synthesis and in situ hybridization of *Tlx* in *P.carnea***. The sequence for *Tlx* transcript was recovered from a newly assembled transcriptome of *P. carnea*. *Tlx* was amplified from medusae cDNA using the following PCR primers: *P. carnea* forward 5'- GAAAGATAAACACGAAAAAGAAACGG-3' and reverse 5'-TCCG GAACTTCATTACTCGCTGTTGC-3' for an expected amplicon length of 528 bp. Amplicons were cloned using the Invitrogen pCR4-TOPO-TA Cloning Kit and sequenced using M13 forward and reverse primers. Sense and antisense DIG labeled riboprobes were synthesized from clones using the Invitrogen T7/T3 Megakrit kit. In situ hybridization (ISH) protocol was adapted from ref. [56]. Only

the fixing solution, the amount of probes and the alkaline phosphatase signal detection solution were changed from the original protocols and are described below. Animals were fixed in ice cold fix (3.7%PFA and 0.25% glutaraldehyde in 1X PBS). Hybridization was carried out at 50 °C for 18 h with a probe concentration of 1 ng/μl. DIG labeled riboprobes localization was detected by immunostaining with anti-DIG-Fab-AP (ROCHE) and NBT/BCIP.

**Probe synthesis and in situ hybridization of _Tlx_ in _P. noctiluca_.** The _PnTlx_ sequence was retrieved from _Pelagia noctiluca_ transcriptome. RNA extraction was performed on five days ephyra using the RNAqueous Total RNA Isolation Kit (Invitrogen AM1912) and cDNA was generated using the reverse transcriptase SuperScript III TM (Invitrogen). PCR products were cloned in pGEM-T Easy plasmid then transformed in competent cells. In situ hybridization probes were synthesized with T7 RNA polymerase. In situ hybridization was performed as previously described[57] with some modifications. Ephyra (5 days post-fertilization), pre-ephyra (3dpf) and planula (2dpf) were generated from fertilized eggs collected from wild jellyfish caught in the bay of Villefranche-sur-Mer (France). Ephyrae were relaxed using 400 μM menthol. Ephyrae, pre-ephyrae and planulae were fixed on ice with a pre-chilled solution of 3.7% formaldehyde plus 0.4% glutaraldehyde in 1X PBS (Phosphate-Buffered Saline), for one hour. Before the acetylation step, embryos were treated for 20 min in 10 μg/ml of proteinase K (Fisher BioReagents™ BP1700-500) followed by two washes in 4 mg/ml of glycine to stop the action of proteinase K. Then embryos were post-fixed in 3.7% formaldehyde in 1X PBST. 5% of dextran was used in the hybridization buffer and SSC pH 4.7.

**De novo transcriptome assembly for _P. carnea_.** For the Podocoryna carnea de novo transcriptome assembly used in this study, planula and early budding polyp libraries were sequenced and the sequences submitted to NCBI Sequence Read Archive (BioProject ID PRJNA744579) in addition to updated libraries for non-reproductive polyps, budding polyps and medusae (BioProject ID PRJNA245897), and used along with previously generated _P. carnea_ libraries described in ref. [14]. Stage 2 planulae (elongated, swimming, non-competent) and stage 1-2 early budding polyps, see Frey[29], were flash frozen and sent to KUMC-GSF for RNA library preparation using the TruSeq RNA Sample Preparation Kit (BoxA). All libraries were 100 bp paired-end with an average insert size of 170 bp. Libraries were then barcoded, pooled, and multiplexed on a single lane of an Illumina NovaSeq 6000 S1 flow cell at KUMC-GSF. Low -quality reads were trimmed and adapters using fastp[52]. Reads from all libraries except the ones from planulae were mapped to the draft genomes of their respective strains of _Podocoryna carnea_ (Chang and Baxevanis, pers. comm.) using STAR[58]. Uniquely mapped reads (61.28%) and reads mapping to multiple loci (20.05%) were kept for de novo assembly. The de novo transcriptome assembly was produced with Trinity v2.8.5[53], yielding 472366 transcripts total with an average length of 714.62 and a G + C content of 36.76%. Transcripts were blasted against a _Mus musculus_ transcriptome dataset (GCA_000001635.9 GRCm39) with an e-value threshold of 1.e^{−100}, and best hits were removed from the assembly. The longest ORFs from the transcriptomes were predicted using Transdecoder v5.5.0 (http://transdecoder.github.io), duplicate sequences and isoforms were removed by clustering sequences with a 95% identity threshold using CD-HIT v4.8.1[59] and a BUSCO analysis[60] against the metazoan database (metazoa_odb10) was performed to assess the completeness of the transcriptome using BUSCO v3.0.2, estimating a 96.2% completeness of the transcriptome (82.9% single copy Buscos and 13.3% duplicated Buscos), 1.5% of fragmented Buscos and 2.3% of missing Buscos on a total of 954 BUSCO markers.

**Differential expression analysis in the life cycle of _P. carnea_, _C. hemisphaerica_ and _A. coerulea_ and _Tlx_ qPCR validation in _P. carnea_.** The differential expression analyses were carried out on the transcriptomes of _Clytia hemisphaerica_ (http://marimba.obs-vlfr.fr;[18]) and _Aurelia coerulea_ (https://davidadlergold.faculty.ucdavis.edu;[61]) and the de novo transcriptome assembly of _Podocoryna carnea_. The reads quantification was performed at the gene level, quantification combining isoforms, using RSEM[62]. Estimated counts, Fragments per Kilobase Million (FPKM) and Transcripts per Million (TPM) values were generated and differential expression of _Tlx_ for the three species was analyzed on these three metrics using Ebseq[63]. The differential expression analysis was performed on the main developmental stages of the life cycle for the three species, planulae (binning the three planulae stages for _Clytia_), non-reproductive polyp (respectively, gastrozooid, scyphostoma and non-reproductive polyp), reproductive polyp (respectively, gonozooid, early and late budding polyps, early and late strobila) and medusa (respectively, ephyra, juvenile and mature). The relative expression of _Tlx_ was validated in _Podocoryna carnea_ through RT-qPCR. Tissues from planulae (7 biological replicates) non-reproductive polyps (8 biological replicates), budding polyps (8 biological replicates) and released medusae (8 biological replicates) were homogenized in Trizol and incubated for 15 min. Samples were then combined with 0.5 volume of chloroform, mixed, incubated for 3 min and then spun down at 4 °C for 15 min. The supernatant was mixed with one volume of 70% ethanol and RNA extraction was carried out using a Qiagen RNAeasy Micro Kit. cDNA synthesis was carried out using Superscript IV and a primer mixture of random hexamers and oligo-dT. cDNA was quantified with Qubit and the qPCR was performed using PowerUP SYBR Green, using the following primers (_Ef1_ forward 5'-TTGCCACCTCAACGACCATC-3', _Ef1_ reverse 5'-TACCGACTGGCACTGTT

CCA-3' and _Tlx_ forward 5'-CAGAGCCCCACCGAAAAGAA-3', _Tlx_ reverse 5'-ATTCCTTGGCCACACGCAAT-3').

**TLX antibody synthesis, Western blot validation and Immunolocalization of TLX in _P. carnea_.** A polypeptide corresponding to _Podocoryna carnea_ TLX 23 amino acids downstream of the EH1 domain (Nter- KDKHEKETELE-KEGKKLSSENIC-Cter) was produced and used to raise antisera in two rabbits by Pacific Immunology Corp. Two rabbits were immunized four times. The whole serum of the two rabbits was extracted, pooled and affinity purified, yielding a pool of polyclonal antibodies (PAC 17963-17964, 2.6 mg/ml).

A His-tagged recombinant polypeptide corresponding to _Podocoryna carnea_ TLX first 114 amino acids (no homeodomain, Nter-MMKKTSLSFSIESILKDKHE KETELEKEGKKLSSENICYYNEAQVNSSKYQLKYFTTVSHHPGIKTVVSFC-GAHF-Cter) was produced by Genscript and used as a control for PAC 17963-17964 validation. 35 ng of TLX recombinant protein (114 aa) and ~10 μg of male medusa protein were electrophoresed at 125 V for 75 min in a 10% gel (Novex 10% Tris-Glycine mini gels (ThermoFisher XP00105BOX). Protein was transferred to a PVDF membrane (Bio-Rad 1620177) using the Pierce G2 Fast Blotter. Blocking was carried out in 5% milk/TBS for 1 hour at room temperature, with rocking, and incubated overnight at 4 C in Ab19763 at 1-2 μg/mL in 10 mL 1 × 5% milk/TBS with rocking. Four post incubation washes were carried out in 10 mL 1xTBST, 15 min each at room temperature with rocking. Secondary antibody incubation (anti-rabbit HRP conjugate, A0545) was carried out at 1:20,000 dilution in TBST for 1 h at room temperature with rocking.Four post incubation washes were carried out in 10 mL 1xTBST, 10 min each at room temperature with rocking. Signal was detected using SuperSignal West Pico Chemiluminescent Substrate kit (ThermoFisher 34,078), exposed to Carestream Biomax XAR Film (Sigma 1,651,454), and developed using a Konica SRX-101A tabletop processor.

The immunostaining protocol was adapted from[64]. Animals were fixed in 3.7% PFA −0.3% PBS-Triton X-100 for an hour at room temperature. Incubation in the primary antibodies was carried out overnight at 4 °C. Samples were incubated in primary antibodies at a 1:1000 dilution in 3% BSA − 0.3% PBS-Triton X-100, using PAC 17963-17964 and Mouse anti-Histone H1 (Leinco Technologies Cat# H126, RRID:AB_2830309). Samples were incubated in secondary antibodies at a 1:1000 dilution in 3% BSA − 5% normal goat serum- 0.3% PBS-Triton X-100, using Goat Anti-Mouse IgG (H + L) secondary antibody conjugated with CF488A (Sigma-Aldrich Cat# SAB4600042, RRID:AB_2532075) and Goat anti-Rabbit IgG (H + L) cross-Adsorbed secondary antibody conjugated with Alexa Fluor 594 (Thermo Fisher Scientific Cat# A-11012, RRID:AB_2534079) for an hour at room temperature. Samples were incubated in 1:1000 Hoechst 33342 (Thermo Fisher Scientific Cat#I34407) − 0.3% PBS-Triton X-100 for 30 minutes at room temperature.

Specimens were imaged on either a Leica DM5000 B compound microscope with a Lumenera INFINITY 3 s camera (Figs. 4b–h and 5a–c) or on a Leica TCS SPE Laser Scanning Confocal upright microscope (Model DM6-Q) (Fig. 5d). Images from different channels were merged using ImageJ v1.53e. Images were adjusted for brightness and contrast; any adjustments were applied to the entire image, not parts.

**Statistics and reproducibility.** Statistical analyses were performed in RStudio v 1.3.1093. Differences between the means were considered significant when $p < 0.05$, using unpaired two-tailed Student's $t$ test or one-way ANOVA. RTqPCR experiments were performed with biological replicates and the individual numerical measurements for each sample can be found in Supplementary Data 1.

Boxplots were generated in R using ggplot2.boxplot. Boxplots the centerline corresponds to the median, the lower and upper hinges correspond to the 1st and 3rd quartile, the upper whisker extends from the hinge to the largest value no further than $1.5 \times IQR$ (inter-quartile range) from the hinge, and the lower whisker extends from the hinge to the smallest value at most $1.5 \times IQR$ of the hinge.

In situ hybridization and Immunostaining analyses were performed on biological replicates to ensure the reproducibility of the data shown. The source of the nucleotide/aminoacid sequences and the alignments used for the various analyses can be found in Supplementary Notes 1–4 and Supplementary Data 1.

**Reporting summary.** Further information on research design is available in the Nature Portfolio Reporting Summary linked to this article.

## Data availability
The authors declare that all relevant data supporting the findings of this study are available within the manuscript and its Supplementary materials. The alignment used to build the tree in Fig. 1b can be found in Supplementary Note 1. The alignment used to build the tree in Supplementary Fig. S1 can be found in Supplementary Note 2. The alignment used to build the tree in Supplementary Fig. S2 can be found in Supplementary Note 3. The source of the genome assemblies, transcriptome assemblies or SRA used to assemble the datasets in Table 1 can be found Supplementary Data 1. The matrix used to perform the Bayesian correlation analysis can be found in Supplementary Data 1. The alignment used to perform the selection analysis in Table 2 can be found in

Supplementary Note 4. The following dataset is publicly available on Genbank: *Podocoryna carnea* SRAs and transcriptome assembly (BioProject ID PRJNA245897).

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

## Acknowledgements

We thank A. Baxevanis and E.S Chang for providing draft genome sequences, A. Klompen for help in data collection. Genome sequencing services were provided by the University of Kansas Medical School Genome Sequencing Center. Computing support was provided by the University of Kansas Center for Research Computing. This work was supported by the National Science Foundation (NSF) Division of Environmental Biology 2153774 the University of Kansas General Research Fund, and the University of Kansas Research Excellence Initiative (to P.C.) and support to P.C. while serving at the National Science Foundation. M.T. was supported by the Chancellor's Doctoral Fellowship at the University of Kansas and received research support from the KU Graduate Research Fund. L.L. and M.B. were supported by the CNRS 80|Prime CNRS program and the Agence Nationale de la Recherche (ANR-13-PDOC-0016).

## Author contributions

P.C., S.S., R.S., M.T. designed research; R.B., M.B., S.M.S. and M.T. performed research; L.L., M.B., S.M.S. and M.T. contributed new reagents/analytic tools; P.C., L.L. and M.T. analyzed data; P.C. and M.T. wrote the paper with input from the other authors.

## Competing interests

The authors declare no competing interests.
