## [Peer Review File · Communications Biology]

Reviewers' comments:

Reviewer #1 (Remarks to the Author):

Travert et al. studied Tlx genes in cnidarians and some other metazoans. They show that the presence of Tlx genes in cnidarian genomes correlates with the presence of a medusa stage in their life cycle. The medusa (=jellyfish) stage is a (mostly) pelagic stage that performs the sexual reproduction task in the life cycles of many medusozoans. A medusa stage is lacking altogether in anthozoans and endocrinozoans but is present in many members of the cnidarian subphylum Medusozoa. Previous work by the Cartwright lab has shown multiple gains and losses of the medusa stage in hydrozoans – the most diverse medusozoan class. Results of mRNA in situ hybridization in *Podocoryna* are consistent with a role of Tlx in medusa development.

The correlation between the presence of a functional Tlx gene and a medusa stage is compelling. It suggests that Tlx fulfills a fundamental role in medusa somatic development. I look forward to seeing the phenotype of Tlx knockout in one of the medusa-bearing hydrozoans that are amenable to genetic manipulation. This is, however, not within the scope of the present study.

This work raises many questions on the modes of character gain and loss in evolution. The PIs make some careful suggestions that can only be tested by additional species sample and functional studies. Even the non-exhaustive sample of the present study makes it quite unlikely that Tlx distribution is unlinked with medusa development. I will be excited to see this work published.

Minor comments:

1. In line 14, the authors refer to Anthozoa as subphylum (which I believe to be correct) but in line 33 they demoted it to a class.
2. The in situ hybridization images are difficult to interpret in the absence of corresponding sense probes (in line 390 they said that a sense probe had been synthesized).
3. Many *Millepora* species actually do have a medusa stage. Is *M. squarrosa* an exception? Have the authors investigated medusa-bearing *Millepora* spp.?

Reviewer #2 (Remarks to the Author):

The manuscript of Travert et al. shows a correlation between the phyletic distribution of the presence/absence of the Tlx gene and the presence/absence of a medusa stage in cnidarians. It addresses thereby two fundamental questions of cnidarian biology: the evolutionary origin on the metagenetic life cycle and the developmental basis for the evolution and secondary loss of the medusa stage. While the correlation between the Tlx gene distribution and the medusa stage is strong and compelling, the lack of comparative expression data does not currently allow assessing if the observed correlation is coincidental or due to a putative conserved role of Tlx during medusa development.

The Tlx gene was studied here in a large number of cnidarian species covering all branches of the cnidarian phylum using transcriptomic and/or genomic resources and confirmed by degenerate PCR on genomic DNA. The authors found a strong correlation between the presence/absence of the Tlx gene and the presence, reduction or absence of a medusa stage. In the few cases where a Tlx gene was found despite the reduction or absence of a medusa stage, the authors convincingly show that these genes are likely not functional and/or under reduced selection pressure. Stage-specific transcriptomes show relative Tlx gene expression levels increase at medusa stages, supporting a role during medusa development. Tlx gene expression patterns are only shown for the hydrozoan *Podocoryne carnea*.

The strength of the paper lies in the exhaustive sampling of cnidarian species. In my opinion, the manuscript's main weaknesses is the questionable quality of the in situ hybridisation data and the

lack of comparative spatial expression data from at least one other hydrozoan or scyphozoan species. This hampers building a hypothesis to explain why Tlx loss is correlated with the primary absence or secondary reduction or loss of the medusa stage.

Major points:

1) The expression patterns of Podocoryne Tlx are partly incompatible with the stage-specific transcriptome results. The transcriptome shows a major increase in Tlx expression levels in budding polyps. That particular stage remains unstained in the in situ hybridisation experiment (Fig. 4B). This contradiction needs to be resolved by detecting Tlx in budding polyps or addressed in the manuscript.

I have general doubts about the medusa expression patterns of Tlx. It is difficult to judge if the stained spherical structures in the gonad region (Fig. 4D and E) is specific or unspecific staining (e.g. food vesicles). Also, it seems that some tentacle bulbs in Fig.4C (e.g. labeled 'tb' and maybe others) are not stained. This suggests that Tlx-expressing cells are only present in a subset of tentacle bulbs, which would be highly unusual or again reflects unspecific staining. Testing for unspecific staining should be addressed using a sense probe control. Also, have the authors starved the animals before fixation? Another source of potential background could be the unusual low hybridisation temperature (50°C). Although no information was available on other protocol parameters (formamide concentration, SSC concentrations), it is unusually low under standard conditions and thus increases the probability of unspecific staining.

The picture quality of the oocyte (Fig.4 F) seems to have been taken using very high phase contrast and needs improvement before publication.

A major problem to interpret the correlation between Tlx distribution and the presence of a medusa is the lack of comparative spatial expression data. As some of the authors have access to the full life cycle of *Clytia hemisphaerica*, the authors should study Tlx expression during *Clytia* medusa development. A comparison between Podocoryne and *Clytia* should inform about the also help evaluating the validity of the Podocoryne pattern.

Also, it would be great to have comparative expression data from a scyphozoan species such as strobilating stages of *Aurelia aurita* to learn about the cells and developmental processes that Tlx might be involved in during medusal development throughout medusozoans.

I believe that spatial expression data from at least one additional species is necessary to evaluate a causality between Tlx loss and medusa reduction or loss.

2) The manuscript mentions the previous reporting of a Tlx gene in *Nematostella* (Ryan et al 2006) and claims that missing EH1 and N-terminal arm domains, and low bootstrap support values do not support its Tlx gene orthology.

A recent BiorXiv preprint (Zimmermann et al. 2020; <https://doi.org/10.1101/2020.10.30.359448>) reports a chromosome-level genome assembly of *Nematostella*. It confirms the presence of a 'Tlx-like gene' in an NK cluster between Ladybird and Hex, a position where Tlx is also found in *Saccoglossus* or amphioxus. The conserved genomic synteny of the 'Tlx-like' gene thus strongly supports the existence of a (degenerate?) Tlx ortholog in *Nematostella* and questions the 'absence' of Tlx as presented here.

As the authors include Tlx genes from a much higher diversity of species compared to the Ryan et al. 2006 paper, I suggest that the authors include the *Nematostella* 'Tlx-like' in their phylogenetic analysis (Fig. S1). Also, the authors should clarify/comment on the proposed conserved syntenic position of the 'Tlx-like' gene in *Nematostella* and its assumed orthology to bilaterian Tlx. They should also recheck other anthozoan genomes (*Acropora*, *Aiptasia*,...) for Tlx-like genes/pseudogenes in the respective genomic regions. It would be interesting to know if in medusozoans with sequenced genomes, Tlx is found in the genome in a similar syntenic context.

Minor points:

- In the introduction (lines 35-37), the authors take a strong stand that medusae are not ancestral to Cnidaria (and Anthozoa). In the discussion, however, two of their hypotheses (#1 and #3, line 290-92) postulate that medusa are ancestral in cnidarians. This is somehow contradictory and it is unclear if this change was driven by the author's Tlx data or was previously already supported in the field.

- As shown in Fig. 1A, the 'signature motif' presented in line 75 is variable at some positions among medusozoan species. The variation on these positions should be reflected in the text. Also, it stays unclear how variable this sequence is between Tlx genes of cnidarians and bilaterians. It would therefore be helpful to add some bilaterian sequences and species names to the alignment of Fig.1A.

- It's unclear if '188 cnidarian transcriptomes' (l.128) refers to transcriptomes of 188 species, or if some species were represented by >1 transcriptome. Similarly (l.119ff) for the '73 ... cnidarian draft genomes' (e.g. there are genomes from different *Aiptasia* strains). The authors should specify how many species were covered.

- While looking in Fig. 2 for the phyletic position of the four non-medusa-bearing species that have retained a Tlx sequence, 2 species (*M. squarrosa*, *D. pumila*) could not be found. If they are truly absent (and were not overlooked by me), could the authors please add them or comment on the reason for their absence?

- There is a mixup in the naming of panels in the figure legend of Fig. 4 (f.ex. D) is not a 'higher magnification of A))

- While the discussion of the evolution of the Tlx gene is well written, the discussion about its assumed function is vague and often trivial. F.ex. in lines 275-278: 'Area of active somatic development' can in principle relate to the entire medusa except the germline, so it is not informative. Also, suggesting a 'role in the maintenance of somatic development' is trivial. Hopefully, with the addition of expression data from at least one additional medusozoan species (as suggested above), this part of the discussion will become more comparative and informative.

- The discussion lays out three scenarios of Tlx/medusa co-evolution in cnidarians. The third scenario (Tlx has ancestral function in development of medusa structures in a cnidarian ancestor) seems fully compatible with the first scenario (suggests possibility of an ancestral metagenic life cycle in Cnidaria). Maybe I am missing something, but is scenario #3 a more elaborate version of #1? If not, the authors need to be more precise in laying out the differences.

Reviewer #1 (Remarks to the Author):

Travert et al. studied Tlx genes in cnidarians and some other metazoans. They show that the presence of Tlx genes in cnidarian genomes correlates with the presence of a medusa stage in their life cycle. The medusa (=jellyfish) stage is a (mostly) pelagic stage that performs the sexual reproduction task in the life cycles of many medusozoans. A medusa stage is lacking altogether in anthozoans and endocrinozoans but is present in many members of the cnidarian subphylum Medusozoa. Previous work by the Cartwright lab has shown multiple gains and losses of the medusa stage in hydrozoans – the most diverse medusozoan class. Results of mRNA in situ hybridization in Podocoryna are consistent with a role of Tlx in medusa development.

The correlation between the presence of a functional Tlx gene and a medusa stage is compelling. It suggests that Tlx fulfills a fundamental role in medusa somatic development. I look forward to seeing the phenotype of Tlx knockout in one of the medusa-bearing hydrozoans that are amenable to genetic manipulation. This is, however, not within the scope of the present study.

This work raises many questions on the modes of character gain and loss in evolution. The PIs make some careful suggestions that can only be tested by additional species sample and functional studies. Even the non-exhaustive sample of the present study makes it quite unlikely that Tlx distribution is unlinked with medusa development. I will be excited to see this work published.

Minor comments:

1. In line 14, the authors refer to Anthozoa as subphylum (which I believe to be correct) but in line 33 they demoted it to a class.

This has been corrected and now Anthozoa is consistently referred to subphylum throughout.

2. The in situ hybridization images are difficult to interpret in the absence of corresponding sense probes (in line 390 they said that a sense probe had been synthesized).

We thank the reviewer for this comment. We have now added a figure in the supplementary section (Figure S5) showing the results of the ISH using the sense probes. No non-specific staining was detected using the sense probes except some non-specific signal was found in some eggs in the female medusae (Figure S5D). This non-specific signal is addressed in the manuscript and specificity of TLX expression in the eggs has been confirmed via immunostaining of TLX. Non-specific blue staining is also found in the skeletal material of the stolon.

3. Many Millepora species actually do have a medusa stage. Is *M. squarrosa* an exception? Have the authors investigated medusa-bearing Millepora spp.?

This is an important point as there is indeed quite a bit of confusion in the literature. The medusoid stage of *Millepora complanata* has mistakenly been described as a

eumedusoid in Lewis,1991. Despite illustrating radial canals, the author claimed to not have actually observed them in the specimens. It should be noted that the original description (Hickson 1899) described it as a cryptomedusoid. In addition other descriptions of several Millepora species by Hickson, 1899, Mayer, 1910, Boschma,1956, Peterson1990 and more recently Amaral et al, 2008, Bourmaud et al, 2013 and Dubé et al 2019 are consistent with these species bearing cryptomedusoids. To add to the confusion, some authors loosely refer to the gonophore type as a medusa when it is in actuality a medusoid. There has never been a description of the gonophore type in *M. squarrosa*, but given the consistency throughout the genus and the monographs from Mayer 1910 and Petersen 1990 regarding the family Milliporidae, we interpret it as possessing a cryptomedusoid stage.

Reviewer #2 (Remarks to the Author):

The manuscript of Traver et al. shows a correlation between the phyletic distribution of the presence/absence of the Tlx gene and the presence/absence of a medusa stage in cnidarians. It addresses thereby two fundamental questions of cnidarian biology: the evolutionary origin on the metagenetic life cycle and the developmental basis for the evolution and secondary loss of the medusa stage. While the correlation between the Tlx gene distribution and the medusa stage is strong and compelling, the lack of comparative expression data does not currently allow assessing if the observed correlation is coincidental or due to a putative conserved role of Tlx during medusa development.

The Tlx gene was studied here in a large number of cnidarian species covering all branches of the cnidarian phylum using transcriptomic and/or genomic resources and confirmed by degenerate PCR on genomic DNA. The authors found a strong correlation between the presence/absence of the Tlx gene and the presence, reduction or absence of a medusa stage. In the few cases where a Tlx gene was found despite the reduction or absence of a medusa stage, the authors convincingly show that these genes are likely not functional and/or under reduced selection pressure. Stage-specific transcriptomes show relative Tlx gene expression levels increase at medusa stages, supporting a role during medusa development. Tlx gene expression patterns are only shown for the hydrozoan *Podocoryne carnea*.

The strength of the paper lies in the exhaustive sampling of cnidarian species. In my opinion, the manuscript's main weaknesses is the questionable quality of the in situ hybridisation data and the lack of comparative spatial expression data from at least one other hydrozoan or scyphozoan species. This hampers building a hypothesis to explain why Tlx loss is correlated with the primary absence or secondary reduction or loss of the medusa stage.

In response to the suggestion, we:

Repeated the in situ hybridization experiments and refined the timing of development to optimize signal. The new in situs presented here are much improved and the expression patterns are clearer.

1. We generated a polyclonal antibody against the TLX protein to provide a more sensitive detection method (see below) and found that these patterns were largely consistent with the *in situ* patterns.
2. We added another species for comparison, the scyphomedusa *Pelagia nocticula*. The ISH for *P. noctiluca* shows that *Tlx* is expressed during medusa development in scyphomedusa specific structures (Fig. 4G-J).

Major points:

The expression patterns of Podocoryne *Tlx* are partly incompatible with the stage-specific transcriptome results. The transcriptome shows a major increase in *Tlx* expression levels in budding polyps. That particular stage remains unstained in the *in situ* hybridisation experiment (Fig. 4B). This contradiction needs to be resolved by detecting *Tlx* in budding polyps or addressed in the manuscript.

Although *in situ* hybridization experiments are not quantitative and thus do not always correlate with RNA-Seq data, we agree with the reviewer that the lack of detected *Tlx* staining in medusa buds was troubling. In repeating the ISH, we were able to prolong the timing of development in the polyps, which in turn revealed *Tlx* staining in the medusa buds of the polyp (Figure 4b). In addition, to confirm these results, we generated polyclonal antibodies against the TLX protein and performed immunostaining (IHC) targeting *P. carnea* TLX. We found a consistent pattern was documented between ISH and IHC including in the medusae buds (see figures 4 and 5).

The modified text is shown below:

Line 243-247: Expression in *P. carnea* was detected in medusa buds and free-living medusae (Figure 4B-F), but not in polyps (Figure 4A), or planulae (not shown). *Tlx* was detected in the distal portion of the medusa bud (stages 3-4) in both the endoderm and ectoderm as well as the cellular membrane surrounding the bud (Figure 4B). *PcTlx* was not detected in early buds (stages 1-2) or in late buds (stages 5-6)

Line 264-267: The non-reproductive polyps, early medusae (stage 1), and later budding stages (5 and above) do not show any TLX expression (Figure 5A, C). TLX expression is observed in the distal portion of medusa of stage 3 buds (Figure 5B), consistent with *in situ* hybridization patterns,

I have general doubts about the medusa expression patterns of *Tlx*. It is difficult to judge if the stained spherical structures in the gonad region (Fig. 4D and E) is specific or unspecific staining (e.g. food vesicles).

We have included the sense probe control (Figure S5) to show that it is not non-specific staining such as food vesicles.

Also, it seems that some tentacle bulbs in Fig.4C (e.g. labeled 'tb' and maybe others) are not stained. This suggests that *Tlx*-expressing cells are only present in a subset of tentacle bulbs, which would be highly unusual or again reflects unspecific staining.

Tlx staining is consistent throughout all the tentacle bulbs, although we agree with the reviewer that it is difficult to see given that tentacle bulbs are not all in the same focal plane. This expression is clearer in the IHC staining in Figure 6 and S4 and should help clarify.

Testing for unspecific staining should be addressed using a sense probe control.

The sense probe control is now shown in Figure S5. No staining was detected except for some in the developing oocytes and in the non-cellular skeletal material.

Also, have the authors starved the animals before fixation?

The colonies were starved four days prior to the collection of tissues and medusae are not fed. This is outlined in the methods section under Animal Care:

Line 470-471: Unfed one and three day old released medusae were collected. Prior to every experiment, *P.carnea* colonies were starved for four days.

Another source of potential background could be the unusual low hybridisation temperature (50°C). Although no information was available on other protocol parameters (formamide concentration, SSC concentrations), it is unusually low under standard conditions and thus increases the probability of unspecific staining.

The protocol is adapted from Gajewski et al 1996, variations from the parameters of the protocol have been clarified in the revised manuscript:

Line 480-485: Only the fixing solution, the amount of probes and the alkaline phosphatase signal detection solution were changed from the original protocols and are described below. Animals were fixed in ice cold fix (3.7%PFA and 0.25% glutaraldehyde in 1X PBS). Hybridization was carried out at 50°C for 18 hours with a probe concentration of 1 ng/µl. DIG labeled riboprobes localization was detected by immunostaining with anti-DIG-Fab-AP (ROCHE) and NBT/BCIP.

We agree with the reviewer that 50C is a low temperature relative to what is typically used in other species. However, it has been found by us in prior publications (Sanders and Cartwright, 2015a, Sanders and Cartwright 2015b and Sanders et al,2020) as well as other authors (Muller et al,1999 and Spring et al 2000) that 50C hybridization temperature is actually optimal for most probes in this particular species. Including the sense probe controls (Fig S5) hopefully alleviates this concern.

The picture quality of the oocyte (Fig.4 F) seems to have been taken using very high phase contrast and needs improvement before publication.

We have retaken this picture and hopefully you will find in much improved (Figure 4F).

A major problem to interpret the correlation between Tlx distribution and the presence of a medusa is the lack of comparative spatial expression data. As some of the authors have access to the full life cycle of *Clytia hemisphaerica*, the authors should study Tlx expression during *Clytia* medusa development. A comparison between *Podocoryne* and *Clytia* should inform about

the also help evaluating the validity of the Podocoryne pattern.

Also, it would be great to have comparative expression data from a scyphozoan species such as strobilating stages of *Aurelia aurita* to learn about the cells and developmental processes that *Tlx* might be involved in during medusal development throughout medusozoans.

I believe that spatial expression data from at least one additional species is necessary to evaluate a causality between *Tlx* loss and medusa reduction or loss.

In response to this comment, we performed ISH on the scyphomedusa *Pelagia nocticula*. Despite our best efforts *Tlx* could not be detected via in situ hybridization in *Clytia hemisphaerica*, likely due to the narrow window of development in which it is expressed. We also tried in *Aurelia* and we could not see the signal because of the large amount background staining. Our in situs in *P. nocticula* are consistent with it being involved in medusa development and are included in the results section and pasted below:

Line 342-351: In *P. noctiluca* ephyrae *Tlx* is expressed broadly in territories undergoing patterning, notably the developing rhopalia. The expression of *Tlx* in the proximal portion of the rhopalar canal suggests a similar role in cell differentiation as in *P. carnea*. Akin to the tentacle bulb of hydromedusae, the proximal-most portion of the rhopalium is characterized by a dense nervous system (Nakanishi et al., 2009) and active cell proliferation. Although no expression patterns were documented in reproductive medusa of *Pelagia* due to its large size, *Tlx* expression in the reproductive medusa of *Podocoryna* was found to be localized to the manubrium, both in cell populations distinct from the germline as well as in mature oocytes in the medusae. Together these results suggest that *Tlx* plays a role in the early patterning of the developing medusa, then in cellular differentiation, and an additional role in oocyte maturation or as a maternal effect gene.

2) The manuscript mentions the previous reporting of a *Tlx* gene in *Nematostella* (Ryan et al 2006) and claims that missing EH1 and N-terminal arm domains, and low bootstrap support values do not support its *Tlx* gene orthology.

A recent BiorXiv preprint (Zimmermann et al. 2020; <https://doi.org/10.1101/2020.10.30.359448>) reports a chromosome-level genome assembly of *Nematostella*. It confirms the presence of a 'Tlx-like gene' in an NK cluster between *Ladybird* and *Hex*, a position where *Tlx* is also found in *Saccoglossus* or *amphioxus*. The conserved genomic synteny of the 'Tlx-like' gene thus strongly supports the existence of a (degenerate?) *Tlx* ortholog in *Nematostella* and questions the 'absence' of *Tlx* as presented here.

As the authors include *Tlx* genes from a much higher diversity of species compared to the Ryan et al. 2006 paper, I suggest that the authors include the *Nematostella* 'Tlx-like' in their phylogenetic analysis (Fig. S1). Also, the authors should clarify/comment on the proposed conserved syntenic position of the 'Tlx-like' gene in *Nematostella* and its assumed orthology to bilaterian *Tlx*. They should also recheck other anthozoan genomes (*Acropora*, *Aiptasia*,...) for *Tlx*-like genes/pseudogenes in the respective genomic regions. It would be interesting to know if in medusozoans with sequenced genomes, *Tlx* is found in the genome in a similar syntenic context.

We thank the reviewer for this thoughtful comment. In response we have revisited this issue. In our re-analysis we now feel we have a better understanding of the origin of the Tlx gene.

Although the Zimmermann et al, 2020 study suggested the presence of a Tlx-like gene, they also made clear that they have not confirmed the orthology of this gene in *Nematostella* to *Tlx*.

They state: "The missing two genes, Tlx-like (its orthology with bilaterian Tlx is not fully certain) and Ladybird (Lbx), are found on the same chromosome respectively 8.2Mb and 9.3Mb upstream of NK6."

In addition, at the reviewers suggestion, we checked the annotation of the genome deposited on <https://simrbase.stowers.org/starletseaanemone> and the coordinates provided above and found that actually Lbx, Hex and what they call a non-anterior Hox are located between NK6 and the Tlx-like gene on Chromosome 5.

There is confusion in the literature when it comes to the orthology of Tlx and the Tlx-like genes. As mentioned above the *Tlx*-like gene of *Nematostella* is found upstream of Lbx in the NK-cluster. The Tlx gene of *Branchiostoma floridae* is also found in this position. The Tlx-like gene of *B.floridae* is however between Hex and Lbx. It is worth noting that the ordering of the gene in the cluster varies greatly between the three species.

In addition, Tlx is not found in the position mentioned by the authors in *Saccoglossus kovalevskii*. In the genome used by the authors (PRJNA42857), Lbx , Hex and Tlx-like/vent-like are found on the scaffold Skow_1.1 scaffold16907. Tlx is however found on the scaffold Skow_1.1 scaffold46468.

Although the NK-cluster is relatively conserved, the ordering of the genes within the cluster is not evidence of homology.

However given the similarity of sequence and chromosomal position, we performed a phylogenetic analysis including the Tlx-like gene of *Nematostella*, other anthozoans, and sponges. The tree is shown in Figure S2 and the relationship to *Tlx* is addressed in the results section and pasted below:

Line 84-89: Although Ryan et al (2006) reported a Tlx gene in the sea anemone *Nematostella*, this gene lacks the signature EH1 domain and N-terminal arm and was not recovered in the Tlx orthology group with sufficient support in their analysis. In our phylogenetic analyses, we recovered this sequence as part of a well-supported orthology group (BS=70, PP=0.99) comprising sequences from other anthozoans as well as some hydrozoans and sponges (Figure S2), all of which lack the EH1 domain and N-terminal arm. Given that they fall within the Tlx clade, they are herein referred to as the Tlx-like gene.

In addition, we added the Tlx-like sequence of *Nematostella* to the alignment shown in Figure S3 to better illustrate the missing EH1 and N-terminal arm and also highlight those regions that are conserved with Tlx.

Given this result, we revisited the origin of Tlx and added it to the discussion starting at Line 308 and pasted below:

In our phylogenetic analyses, we recovered a *Tlx-like* orthology group, that lacks the N-terminal arm and EH1 domain, within a well-supported clade containing the cnidarian and bilaterian *Tlx* genes (Figure S2). Given that some cnidarian taxa possess both a *Tlx-like* and *Tlx* gene and that the *Tlx-like* gene is found in sponges, *Tlx* likely emerged following a duplication event from the *Tlx-like* gene. The relative position of ctenophores and sponges remains uncertain, thus the absence of the *Tlx-like* gene in ctenophores could either suggest that the *Tlx-like* gene emerged in the last common ancestor of Metazoa and was lost in ctenophores or emerged in the last common ancestor of a putative clade composed of Porifera, Cnidaria and Bilateria (Figure 6).

Minor points:

- In the introduction (lines 35-37), the authors take a strong stand that medusae are not ancestral to Cnidaria (and Anthozoa).

It was not our intention to take a strong stand about whether or not medusae were ancestral in the introduction. We removed the sentence that suggested that any consensus has been reached on the topic.

In the discussion, however, two of their hypotheses (#1 and #3, line 290-92) postulate that medusa are ancestral in cnidarians. This is somehow contradictory and it is unclear if this change was driven by the author's *Tlx* data or was previously already supported in the field.

It was also not our intention to propose that hypothesis #3 meant that medusae were ancestral. We instead meant to postulate a polyploid like ancestor that had polyp but also some medusa-like characteristics – as is the case in extant staurozoans. We modified this paragraph and added a figure (Figure 6) to summarize and clarify the three scenarios. The modified text is pasted below.

Line 368-374: A third explanation is that *Tlx* had an ancestral function in the development of specific structures in a cnidarian monogenetic ancestor that exhibited both polypoid and medusoid features. In that scenario, the emergence of a discrete polyp stage in anthozoans, reduction of the ancestral body plan in parasitic endocnidozoans, and the uncoupling of these features into two discrete generations in medusozoan could have been responsible for the pattern of loss and maintenance of *Tlx* respectively (Figure 6 green). The presence of the late Ediacarian fossil *Hootia quadriformis*, that exhibits both polypoid and medusoid traits would be consistent with this scenario (Liu et al, 2014).

- As shown in Fig. 1A, the 'signature motif' presented in line 75 is variable at some positions among medusozoan species. The variation on these positions should be reflected in the text. Also, it stays unclear how variable this sequence is between *Tlx* genes of cnidarians and bilaterians. It would therefore be helpful to add some bilaterian sequences and species names to the alignment of Fig.1A.

The variability of the signature motif of the N-terminal arm has been clarified with the following sentence:

Line74-77: This motif knows very little variation; however, the third residue may vary in medusozoans with the following frequencies found in surveyed species, isoleucine (~70%), leucine (~17%) and valine (~12%). In surveyed bilaterians, the third residue was consistently found to be either an isoleucine (~88%) or a valine (~12%). Bilaterians sequences were added to the figure and the name of the species displayed.

- It's unclear if '188 cnidarian transcriptomes' (l.128) refers to transcriptomes of 188 species, or if some species were represented by >1 transcriptome. Similarly (l.119ff) for the '73 ... cnidarian draft genomes' (e.g. there are genomes from different Aiptasia strains). The authors should specify how many species were covered.

The number of species surveyed for both genomes and transcriptomes has been clarified and updated in the text.

Line 119 “In our search for the *Tlx* gene in publicly available cnidarian draft genome assemblies from 81 species”

Line 128 “*Tlx* was also searched in the transcriptomes from 188 cnidarian species”

- While looking in Fig. 2 for the phyletic position of the four non-medusa-bearing species that have retained a *Tlx* sequence, 2 species (*M. squarrosa*, *D. pumila*) could not be found. If they are truly absent (and were not overlooked by me), could the authors please add them or comment on the reason for their absence?

The reviewer is correct that *M. squarrosa*, *D. pumila* are not present on the tree in Figure 2. The tree presented in Figure 2 is a tree pruned from Cartwright and Nawrocki 2010, that does not contain any of those species. *M.squarrosa* and *D. pumila* *Tlx* sequences have been found in transcriptomes.

- There is a mixup in the naming of panels in the figure legend of Fig. 4 (f.ex. D) is not a 'higher magnification of A)

We thank the reviewer for catching this. The labeling has been corrected

- While the discussion of the evolution of the *Tlx* gene is well written, the discussion about its assumed function is vague and often trivial. F.ex. in lines 275-278: 'Area of active somatic development' can in principle relate to the entire medusa except the germline, so it is not informative. Also, suggesting a 'role in the maintenance of somatic development' is trivial. Hopefully, with the addition of expression data from at least one additional medusozoan species (as suggested above), this part of the discussion will become more comparative and informative.

With addition of ISH in *P.noctiluca* and IHC on *P.carnea*, we feel we now have better insight into the potential role *Tlx*. We added a portion in the discussion addressing the putative role of *Tlx* in somatic development with more details, pasted below:

Line 334-351: Although axial patterning of newly released hydromedusae and scyphozoan ephyrae is mostly complete, additional development of tentacles, sense organs and nematocytes (stinging cells) continue. In hydromedusae, the tentacle bulbs serve as regions of active cell proliferation, neurogenesis and nematogenesis (Gröger and Schmid, 2000, Denker et al, 2008, Fujita et al, 2019 and Figure S4 D). In *Podocoryna*, TLX is expressed in those territories and most distinctively in rows of cells proximal to the connection of the radial canal to the bulb (see Figure S4 C) suggesting TLX could be involved in the aforementioned processes.

In *P. noctiluca* ephyrae *Tlx* is expressed broadly in territories undergoing patterning, notably the developing rhopalia. The expression of *Tlx* in the proximal portion of the rhopalar canal suggests a similar role in cell differentiation as in *P. carnea*. Akin to the tentacle bulb of hydromedusae, the proximal-most portion of the rhopalium is characterized by a dense nervous system (Nakanishi et al., 2009) and active cell proliferation. Although no expression patterns were documented in reproductive medusa of *Pelagia* due to its large size, *Tlx* expression in the reproductive medusa of *Podocoryna* was found to be localized to the manubrium, both in cell populations distinct from the germline as well as in mature oocytes in the medusae. Together these results suggest that *Tlx* plays a role in the early patterning of the developing medusa, then in cellular differentiation, and an additional role in oocyte maturation or as a maternal effect gene.

- The discussion lays out three scenarios of *Tlx*/medusa co-evolution in cnidarians. The third scenario (*Tlx* has ancestral function in development of medusa structures in a cnidarian ancestor) seems fully compatible with the first scenario (suggests possibility of an ancestral metagenic life cycle in Cnidaria). Maybe I am missing something, but is scenario #3 a more elaborate version of #1? If not, the authors need to be more precise in laying out the differences.

We intended to suggest that the third scenario is not a metagenetic life cycle, but a polypoid life cycle that has medusa-like characters, which later became separated into two separate life cycles in medusozoans. We re-worked that paragraph, included Figure 6 to better illustrate these different scenarios, as discussed in our comment above.

REVIEWERS' COMMENTS:

Reviewer #2 (Remarks to the Author):

I thank the authors for their considerable effort in adding experiments, analysis and clarifying the manuscript. All my previous concerns are now satisfactorily addressed. I support the publication of this manuscript but suggest a few minor things that authors should fix without further revision:

- Maybe I missed it, but I could not find any information on the type of microscopy (epifluorescence or confocal?) performed on the fluorescent samples. Some information in the figure legend and more detailed descriptions in the materials&methods should be added.
- In the last paragraph on p. 10, there seems to be some mis-referencing of figures (Fig. 55 and 5C mismatch). Also Figure S3 (line 272) does not show TLX staining as indicated in the text but alignments.
- In the same paragraph, several sentences make statements about IHC signal detection without referencing to any specific figure. See f.ex. lines 267 or 270.
- There is a mismatch between the strongly staining medusae bud in Fig. 5B ('b4') and the text ('stage 3 buds', line 266). Please correct.
- The labeling on the x-axis of the bars in Figure 3 are invisible due to very small size.
- It remains unclear to a non-specialist to which subclass or subphylum the species names mentioned on p.5, line 138-9 and in l.158-9 on p.6 belong to. That information is necessary to also understand the reference to Table 1, which contains no species names.

We thank the reviewer for their comments and have made all the suggested changes. These are explained below.

I thank the authors for their considerable effort in adding experiments, analysis and clarifying the manuscript. All my previous concerns are now satisfactorily addressed. I support the publication of this manuscript but suggest a few minor things that authors should fix without further revision:

1.

Maybe I missed it, but I could not find any information on the type of microscopy (epifluorescence or confocal?) performed on the fluorescent samples. Some information in the figure legend and more detailed descriptions in the materials&methods should be added.

We thank the reviewer for this comment. A paragraph about the fluorescence microscopy has been added to the method section.

Line 522-525: Specimens were imaged on either a Leica DM5000 B compound microscope with a Lumenera INFINITY 3s camera or on a Leica TCS SPE Laser Scanning Confocal upright microscope (Model DM6-Q). Images from different channels were merged using ImageJ v1.53e. Images were adjusted for brightness and contrast; any adjustments were applied to the entire image.

2.

In the last paragraph on p. 10, there seems to be some mis-referencing of figures (Fig. 55 and 5C mismatch). Also Figure S3 (line 272) does not show TLX staining as indicated in the text but alignments.

This has been corrected.

The paragraph now refers to the proper figures.

Line 231-243: To detect TLX protein during medusa development in *Podocoryna carnea*, we used polyclonal antiserum generated against a portion of PcTLX. The non-reproductive polyps, early medusae (stage 1), and later budding stages (5 and above) do not show any TLX expression (Figure 5a, 5b). TLX expression is observed in the distal portion of medusa of stage 4 buds (Figure 5b), consistent with *in situ* hybridization patterns. In reproductive polyps TLX was detected in the tentacles of the polyp (Figure 5b). In released medusae, TLX is localized to the tentacle bulbs and manubrium, consistent with observed *in situ* patterns (Figure 5c). In tentacle bulbs, TLX has a scattered expression pattern although more concentrated to the proximal portion of the bulb (Figure 5c, Figure S4c). In the manubrium, TLX is expressed in a scattered pattern throughout the manubrium (Figure 5b). TLX positive cells were sporadically observed in the lumen of radial and circular canals (Figure S4). In female medusae, TLX was detected in the oocytes (Figure 5c, 5d). In all TLX-positive cells, the protein was localized to the cytoplasm, which is unexpected for a transcription factor (Figure 5d and Figure S4b).

3.

In the same paragraph, several sentences make statements about IHC signal detection without referencing to any specific figure. See f.ex. lines 267 or 270.

This has been corrected.

Line 232-236: The non-reproductive polyps, early medusae (stage 1), and later budding stages (5 and above) do not show any TLX expression (Figure 5a, 5b). TLX expression is observed in the distal portion of medusa of stage 4 buds (Figure 5b), consistent with *in situ* hybridization patterns. In reproductive polyps TLX was detected in the tentacles of the polyp (Figure 5b). In released medusae, TLX is localized to the tentacle bulbs and manubrium, consistent with observed *in situ* patterns (Figure 5c).

4.

There is a mismatch between the strongly staining medusae bud in Fig. 5B ('b4') and the text ('stage 3 buds', line 266). Please correct.

This has been corrected.

Line 233-234: TLX expression is observed in the distal portion of medusa of stage 4 buds (Figure 5b)

5.

The labeling on the x-axis of the bars in Figure 3 are invisible due to very small size.

The font size of the x-axes of all the graphs in Figure 3 has been increased.

6. It remains unclear to a non-specialist to which subclass or subphylum the species names mentioned on p.5, line 138-9 and in l.158-9 on p.6 belong to. That information is necessary to also understand the reference to Table 1, which contains no species names.

Information about the taxonomy of the species mentioned has been added to the main text.

Line 122-126: *Millepora squarrosa* (Anthoathecata, Milleporidae) *Tlx* is likely a pseudogene as it exhibits premature stop codons and the *Dynamena pumila* (Leptothecata, Sertulariidae) sequence was a partial sequence and thus the presence of a complete coding sequence is unknown. Although *Ectopleura larynx* (Anthoathecata, Tubulariidae) has the characteristic *Tlx* domains

Line 140-146: The three non-medusa bearing species for which a *Tlx* fragment was recovered were the cryptomedusoid bearing *Ectopleura larynx*, also found in the transcriptome above, as well as two sertulariid species that bear sporosacs (*Amphisbetia minima* and *Sertularia perpusilla*). The sequence from *Sertularia perpusilla* (Leptothecata, Sertulariidae), like the sequence of *Millepora squarrosa* discussed above, is likely a pseudogene as it contains several premature stop codons. Thus, of the total of five TLX sequences isolated from non-medusae bearing species from transcriptomes and/or PCR, only *Amphisbetia minima* (Leptothecata, Sertulariidae) has a typical TLX sequence within the amplified region.